# A Statistical Manifold Framework for Point Cloud Data

## Abstract

A large class of problems in machine learning involve data sets in which each data point is a point cloud in $\mathbb{R}^D$. Many applications involving point cloud data sets require a means of measuring not only distances, but also angles, volumes, derivatives, and other more advanced geometric notions. Existing approaches for the most part tend to be ad hoc, and lack coordinate-invariance. In this paper we develop a Riemannian geometric structure for point cloud data. By interpreting a point cloud as samples from some underlying probability distribution, the space of point cloud data can be given the structure of a statistical manifold, with the Fisher information metric acting as a natural Riemannian metric. The only requirement on the part of the user is the choice of a meaningful underlying probability distribution, which is more intuitive and natural to make than what is required in existing ad hoc formulations. Two autoencoder case studies involving point cloud data are presented to demonstrate the advantages of our statistical manifold framework: (i) interpolating between two 3D point cloud data sets to smoothly deform one object into another; (ii) finding an optimal set of latent space coordinates that minimizes distortion.

## 1 Introduction

A growing number of machine learning problems involve data sets in which each data point is a point cloud in $\mathbb{R}^D$, e.g., 3D point cloud obtained by depth cameras. Typical applications include measuring the degree of similarity betweeen two point clouds – the point clouds may be measurements obtained from a depth camera, for example – for which a distance metric on the space of point clouds is needed. Some widely used distance metrics used in this context include the Hausdorff distance (both the original and averaged versions), the chamfer distance (Hausdorff, 1914), and the earth mover distance (Rubner et al., 2000).

However, the distance metric measures just one aspect of point cloud data; other applications may require more advanced concepts and tools. For example, in the case of a moving point cloud, one may wish to measure some of its more dynamic aspects like its velocity or other quantities that require a notion of higher-order derivatives. Applications such as Monte Carlo sampling may require the construction of an isotropic Gaussian distribution on the underlying space, in which case the notion of "angle" is needed in addition to distance (more technically, an inner product on the tangent space is required (Girolami & Calderhead, 2011; Gemici et al., 2016; Mallasto & Feragen, 2018).

Related to the above, the idea of interpreting a point cloud as samples from some underlying probability distribution is well-known, and has been applied to problems ranging from point set registration (Jian & Vemuri, 2005; Wang et al., 2006; Myronenko & Song, 2010; Hasanbelliu et al., 2014; Zhou et al., 2014; Min et al., 2018; Li et al., 2021) to point cloud de-noising (Zaman et al., 2017; Luo & Hu, 2021). In these approaches, divergence measures from probability and informa-

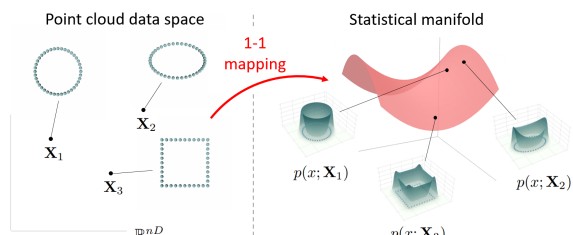

Figure 1: Illustration of statistical manifold obtained from the 1-1 mapping between the set of point cloud data and the space of probability density functions.

tion theory have been utilized to compute the similarity between point clouds. While effective for certain applications, these divergence measures still only capture just one aspect of point cloud data, and cannot be used to measure other quantities of a more geometric nature.

For more advanced applications, a rigorous mathematical characterization of the space of point cloud data is an essential ingredient to a more comprehensive, robust, and correct (in the sense of being coordinate-invariant and geometrically well-defined) analysis of the types described above, particularly one based on **Riemannian geometry**. As such, the first contribution of this paper is a Riemannian geometric structure for the space of point cloud data.

The key idea behind our approach draws upon the information geometry framework of (Amari & Nagaoka, 2000; Amari, 2016), in which the space of probability density functions is given the structure of a Riemannian manifold (the **statistical manifold**), with the Fisher information acting as a natural Riemannian metric (the **Information Riemannian metric**, or info-Riemannian metric for short). The connection between point cloud data and information geometry is established by constructing a 1-1 mapping from the space of point cloud data to the space of probability density functions, i.e., a point cloud $\mathbf{X} = \{x_1, ..., x_n \mid x_i \in \mathbb{R}^D\}$ is mapped to a density function $p(x; \mathbf{X})$ on $\mathbb{R}^D$ in a 1-1 fashion as illustrated in Figure 1.

Two case studies involving autoencoders are presented to demonstrate the benefits of our approach. In the first case study, a pre-trained autoencoder is used to encode two 3D point clouds – one representing a cylinder, one a cone – and the minimal geodesic (or path of shortest length) with respect to the info-Riemannian metric is then constructed between these two objects. The shape evolution obtained for the info-Riemannian metric is seen to be far more natural and intuitive than that obtained for the straight line interpolant in latent space.

In the second case study, we use the info-Riemannian statistical manifold framework to find a set of distortion minimizing latent space coordinates, in the sense that (Euclidean) straight lines in the latent space closely approximate minimal geodesics on the statistical manifold. Such a set of coordinates offers a more discriminative representation for the data manifold (Chen et al., 2020) that results in, e.g., higher linear SVM classification accuracy vis-á-vis existing state-of-the art methods. Experiments are carried out with both synthetic and standard benchmark datasets (*ShapeNet* (Chang et al., 2015), *ModelNet* (Wu et al., 2015)).

## 2 STATISTICAL MANIFOLDS AND THE FISHER INFORMATION METRIC

We begin by extending the original definition of a statistical manifold as follows:

**Definition 1.** *Given an $m$-dimensional topological manifold[1] $\Theta$ and a 1-1 map from $\Theta$ to the space of probability density functions $\theta \mapsto p(x; \theta)$, the image of this mapping, denoted $\mathcal{S} := \{p(x; \theta) | \theta \in \Theta\}$, is an $m$-dimensional statistical manifold.*

In the original definition $\Theta$ is taken to be an open subset of $\mathbb{R}^m$ (Amari & Nagaoka, 2000; Amari, 2016)).

By endowing $\mathcal{S}$ with a Riemannian metric, $\mathcal{S}$ can be given the structure of a Riemannian manifold, allowing for lengths, angles, and volumes to be defined on $\mathcal{S}$ in a coordinate-invariant manner. The Fisher information metric serves as a natural Riemannian metric on $\mathcal{S}$: the elements $(g_{ij})$ of the Fisher information metric $G(\theta) \in \mathbb{R}^{m \times m}$ can be expressed as

$$g_{ij}(\theta) := \int p(x; \theta) \frac{\partial \log p(x; \theta)}{\partial \theta_i} \frac{\partial \log p(x; \theta)}{\partial \theta_j} \, dx, \;\; i, j = 1, \dots, m, \tag{1}$$

where $\theta = (\theta^1, ..., \theta^m)$ are local coordinates on $\mathcal{S}$. Defining infinitesimal length on $\mathcal{S}$ by $ds^2 = d\theta^T G(\theta) d\theta$, the length of a curve $\theta(t)$ on $\mathcal{S}$ can then be computed as the integral $\int_0^T ds$. Further details on statistical manifolds and the Fisher information metric can be found in, e.g., (Amari & Nagaoka, 2000; Efron & Hinkley, 1978; Rissanen, 1996; Han & Park, 2014).

---

[1]A topological manifold is a locally Euclidean Hausdorff topological space.

# 3 STATISTICAL MANIFOLD FRAMEWORK FOR POINT CLOUD DATA

With the above statistical manifold preliminaries, we now construct a Riemannian geometric structure for the space of point cloud data. Section 3.1 defines a statistical manifold from the point cloud data, while Section 3.2 uses the Fisher information metric to construct a Riemannian metric for point cloud data. To keep the definitions and results simple, we shall assume throughout all point cloud data consists of exactly $n$ distinct points in $\mathbb{R}^D$, i.e., each point cloud is of the form $\mathbf{X} = \{x_1, ..., x_n | x_i \in \mathbb{R}^D, x_i \neq x_j \text{ if } i \neq j\}$. The set of all point clouds is denoted $\mathcal{X}$. Later we discuss methods for dealing with point clouds that violate our assumptions. The proofs of propositions in this section are in Appendix B.

## 3.1 STATISTICAL MANIFOLD OF POINT CLOUD DATA

The core idea for constructing the statistical manifold is to interpret a point cloud $\mathbf{X}$ as a set of $n$ samples drawn from some underlying probability density function. Using a kernel density estimator (Parzen, 1962; Davis et al., 2011), a parametric probability density function $p(x; \mathbf{X})$ can be defined in which $\mathbf{X}$ itself is the parameter:

**Definition 2.** *Given a positive kernel function $K : \mathbb{R}^D \to \mathbb{R}$ such that $\int_{\mathbb{R}^D} K(u)\, du = 1$ and a $D \times D$ symmetric positive-definite matrix $\Sigma$ (the bandwidth matrix), the kernel density estimate*

$$p(x; \mathbf{X}) := \frac{1}{n\sqrt{|\Sigma|}} \sum_{i=1}^{n} K(\Sigma^{-\frac{1}{2}}(x - x_i)) \tag{2}$$

*is said to be a statistical representation of the point cloud $\mathbf{X} \in \mathcal{X}$. The set of statistical representations is denoted $\mathcal{S} := \{p(x; \mathbf{X}) \mid \mathbf{X} \in \mathcal{X}\}$.*

To ensure that $\mathcal{S}$ is a statistical manifold, recall from Definition 1 that the following two conditions need to be satisfied: (i) $\mathcal{X}$ is a topological manifold; (ii) A 1-1 mapping $h : \mathcal{X} \to \mathcal{S}$, $\mathbf{X} \mapsto p(x; \mathbf{X})$ must be defined. The first condition can be satisfied with the "distinct points" assumption:

**Proposition 1** (Corollary 2.2.11. in (Knudsen, 2018)). *The set of point clouds in which each point cloud is a set of $n$ distinct points of dimension $D$, is an $nD$-dimensional topological manifold.*

To satisfy the second condition, additional assumptions are needed. The following proposition provides a sufficient condition for $h$ to be 1-1:

**Proposition 2.** *If the set of functions $\{K(\Sigma^{-\frac{1}{2}}(x - x_i)) | x_i \in \mathcal{F}\}$ are linearly independent[2] for any arbitrary finite subset $\mathcal{F} \subset \mathbb{R}^D$ with $|\mathcal{F}| \leq 2n$, the mapping $h : \mathcal{X} \to \mathcal{S}$ is 1-1.*

With the above proposition, any kernel function that satisfies the linear independence condition is sufficient to ensure the existence of a 1-1 mapping $h$. For our purposes the standard and widely used normal kernel function satisfies the linear independence condition.

**Proposition 3.** *Under the distinct points assumption, the standard (multivariate) normal kernel function*

$$K(u) = \frac{1}{\sqrt{(2\pi)^D}} \exp(-\frac{u^T u}{2}) \tag{3}$$

*with the scaled identity bandwidth matrix, i.e., $\Sigma = \sigma^2 I$, satisfies the linear independence condition in Proposition 2.*

From Propositions 2 and 3 we have established that, under the distinct points assumption and using the standard normal kernel function, the mapping $h : \mathcal{X} \to \mathcal{S}$ is 1-1; $\mathcal{S}$ can therefore be given the structure of statistical manifold. While other choices of kernel function are possible, throughout the remainder of the paper we use the standard normal kernel function. Figure 2 illustrates statistical manifold representations of some example point clouds.

---

[2]The linear independence of a set of functions implies that only a trivial linear combination of the functions equals the zero function.

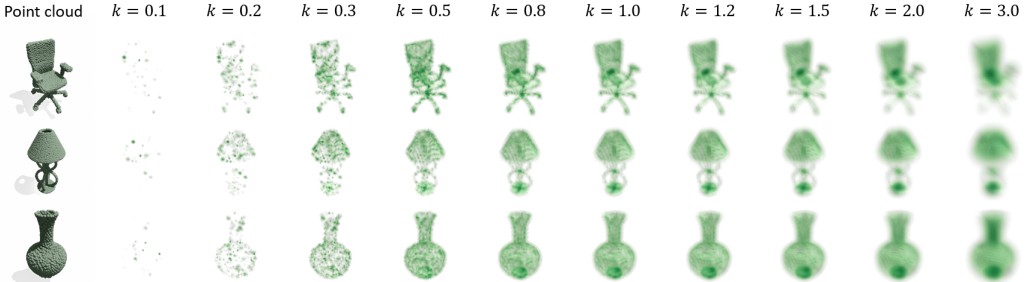

Figure 2: Probability heat maps for various $k$ (the greener, the higher) for some examples from the ShapeNet dataset (Chang et al., 2015), where we set $\sigma = k \times$ MED for $k \in (0, \infty)$. MED denotes the median of the distances between the points in the point cloud and their nearest points.

## 3.2 INFORMATION RIEMANNIAN METRIC FOR POINT CLOUD DATA SPACE

We now equip the point cloud statistical manifold $\mathcal{S}$ with the Fisher information metric, which we refer to as the info-Riemannian metric and denote by $\mathbf{H}$. The first task is to define a local coordinate system on the space of point clouds $\mathcal{X}$. Toward this end, we use the matrix representation $X \in \mathbb{R}^{n \times D}$ of a point cloud $\mathbf{X}$. Observe that the matrix representation is not unique; given an $n \times n$ permutation matrix $P \in \mathbb{R}^{n \times n}$, then $X$ and $PX$ represent the same point cloud $\mathbf{X}$. Fortunately, this does not cause problems since $p(x; X)$ is defined in a permutation-invariant way, i.e., $p(x; X) = p(x; PX)$ for any $n \times n$ permutation matrix $P$. Throughout we use italics to denote local coordinate representations, e.g., $\mathbf{X}$ has local coordinates $X \in \mathbb{R}^{n \times D}$, the tangent vector $\mathbf{V} \in T_{\mathbf{X}}\mathcal{X}$ has local coordinates $V \in \mathbb{R}^{n \times D}$.

The info-Riemannian metric $\mathbf{H}$ can be expressed in local coordinates coordinates $X$ as follows:

$$H_{ijkl}(X) := \int p(x; X) \frac{\partial \log p(x; X)}{\partial X^{ij}} \frac{\partial \log p(x; X)}{\partial X^{kl}} \, dx, \tag{4}$$

for $i, k = 1, ..., n$ and $j, l = 1, ..., D$. Given two tangent vectors $\mathbf{V}, \mathbf{W} \in T_{\mathbf{X}}\mathcal{X}$ with respective matrix representations $V, W \in \mathbb{R}^{n \times D}$, their inner product is then computed as follows:

$$\langle \mathbf{V}, \mathbf{W} \rangle_{\mathbf{X}} := \sum_{i,k=1}^{n} \sum_{j,l=1}^{D} H_{ijkl}(X) V^{ij} W^{kl}. \tag{5}$$

We note that the coordinate expression of the info-Riemannian metric $H_{ijkl}(X)$ results in a permutation-invariant inner product, i.e., $\sum H_{ijkl}(X) V^{ij} W^{kl} = \sum H_{ijkl}(PX)(PV)^{ij}(PW)^{kl}$ for any $n \times n$ permutation matrix $P$, showing that the metric is geometrically well-defined.

Using the standard normal kernel function, the coordinate expression of the info-Riemannian metric $H_{ijkl}$ has a simple analytic expression as follows:

**Proposition 4.** *With the standard (multivariate) normal kernel function and the bandwidth parameter $\sigma$, the information Riemannian metric is given by*

$$H_{ijkl}(X) = \int p(x; X) \frac{K\left(\frac{x-x_i}{\sigma}\right) K\left(\frac{x-x_k}{\sigma}\right)}{\left(\sum_{m=1}^{n} K\left(\frac{x-x_m}{\sigma}\right)\right)^2} \left[ \frac{(x - x_i)(x - x_k)^T}{\sigma^4} \right]_{jl} dx, \tag{6}$$

Figure 3 shows that, given two moving point cloud data whose velocity matrices have equal Euclidean norm (i.e., $\|\mathbf{V}\|^2 = \sum_{i=1}^{n} \sum_{j=1}^{D} V^{ij} V^{ij}$), the velocity norms under the info-Riemannian metric are significantly different: the velocity A has a value of $0.2626$, while the velocity B has a value of $2.2 \times 10^{-8}$. In particular, observe that the tangential velocity in the case B, which does not change the overall distribution of the point cloud, has a very small velocity norm under the info-Riemannian metric as it should.

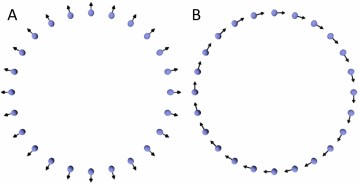

Figure 3: Two moving point clouds with different velocity matrices.

Figure 4: Random walks for point cloud data on the statistical manifold equipped with Euclidean metric (*upper*) and info-Riemannian metric (*lower*). An initial shape is defined by slightly deforming the reference sphere with the addition of a small velocity matrix $V$. Then, sequences of point clouds are generated by recursively adding randomly sampled velocity matrices normalized to have the same norm with $V$ under each metric. Details of implementations are in Appendix C.

As another illustrative example that highlights the difference between the info-Riemannian metric and Euclidean metric (i.e., $\langle \mathbf{V}, \mathbf{W} \rangle := \sum_{i=1}^{n} \sum_{j=1}^{D} V^{ij} W^{ij}$), Figure 4 shows random walks for point cloud data under these metrics. Consider a randomly sampled and normalized velocity matrix under Euclidean metric. 3D velocity vectors in the sampled velocity matrix are equally likely to point any direction. On the other hand, 3D velocity vectors in the sampled velocity matrix under info-Riemannian metric are more likely to point tangential directions (e.g., more likely the case B than the case A in Figure 3). As a result, unlike the standard Euclidean metric (top), the info-Riemannian metric (bottom) produces random walks that stay close to the initial sphere without significant changes in its overall distribution pattern.

## 4 APPLICATIONS TO POINT CLOUD AUTOENCODERS

Riemannian geometric formulations of autoencoders for representation learning have recently been introduced and extensively studied in (Shao et al., 2018; Arvanitidis et al., 2018; Yang et al., 2018a; Chen et al., 2018; Kalatzis et al., 2020; Arvanitidis et al., 2020; Chen et al., 2020). In these works, the image of the decoder function is viewed as a low-dimensional manifold embedded in the high-dimensional data space – we refer to this manifold as the *decoded manifold* – and a Riemannian metric for the decoded manifold is obtained by projecting the data space Riemannian metric to this manifold. In contrast, this perspective cannot be reasonably extended to existing point cloud autoencoders (e.g., FoldingNet (Yang et al., 2018b), AtlasNet (Groueix et al., 2018), AtlasNetV2 (Deprelle et al., 2019), and TearingNet (Pang et al., 2021)), due to the absence of a geometrically well-formulated Riemannian manifold structure.

In this section, using the info-Riemannian metric, we extend this perspective by defining a Riemannian metric for the decoded manifold of the point cloud autoencoder. With this info-Riemannian metric, we examine two case studies: (i) interpolation between two points of latent space via the minimal geodesic; (ii) learning an optimal set of latent space coordinates that best preserves distances and angles (or intuitively, minimizes distortion).

Consider a point cloud decoder function with the $m$-dimensional latent space $f : \mathbb{R}^m \to \mathbb{R}^{n \times D}$, where the output is expressed in terms of the matrix representation. The projection of the info-Riemannian metric on the point cloud statistical manifold to the decoded manifold is then expressed in latent space coordinates $z \in \mathbb{R}^m$ as follows:

$$G_{ab}(z) := \sum_{i,k=1}^{n} \sum_{j,l=1}^{D} H_{ijkl}(f(z))(J_f)_a^{ij}(z)(J_f)_b^{kl}(z), \tag{7}$$

where $J_f$ denotes the Jacobian of $f$, and the indices $a, b$ both range from 1 to $m$. The latent space is then assigned a Riemannian metric $G_{ab}(z)$; the following two applications rely on $G(z) \in \mathbb{R}^{m \times m}$.

**Geodesic interpolation:** The latent space metric $G(z)$ can be used to find the minimal geodesic curve connecting two point clouds (i.e., the shortest length curve in the decoded manifold). Let $z_1, z_2$ be the encoded values of the two point clouds in the latent space $\mathbb{R}^m$. In terms of the metric $G(z)$, the geodesic curve connecting these two points can be determined as a solution to the following optimization problem (Do Carmo, 2016):

$$\min_{z(t)} \int_0^1 \dot{z}(t)^T G(z(t)) \dot{z}(t) \, dt, \tag{8}$$

subject to $z(0) = z_1$ and $z(1) = z_2$. Parametrizing $z(t)$ by a cubic spline with fixed boundary points $z_1, z_2$ then leads to an unconstrained optimization problem. To avoid excessive memory consumption when computing the objective function and its gradient, instead of the usual Riemann sum approximation of the integral, we interpret the integral as an expectation over the uniform distribution $t \sim U(0, 1)$ and accordingly use the mini-batch sampling technique.

**Learning optimal latent space coordinates:** The latent space metric $G(z)$ can be used to formulate a regularization term when training an autoencoder to learn an optimal set of latent space coordinates; by "optimal" we mean $G(z) = cI$ for some positive scalar $c$, so that the decoder preserves distances and angles as much as possible. Recently, a regularization technique for this purpose has been introduced in (Chen et al., 2020). Specifically, the following regularization term is added to the reconstruction loss function:

$$\mathbb{E}_{z \sim P}[\|G(z) - cI\|_F^2], \qquad (9)$$

where $\| \cdot \|_F$ is the Frobenius norm, $c = \mathbb{E}_{z \sim P}[\frac{1}{m}\text{Tr}(G(z))]$, and $P$ is defined via the modified mix-up augmentation, i.e., $z \sim P \iff z = \alpha z_1 + (1 - \alpha)z_2$ where $z_1, z_2$ are sampled from the set of encoded training data and $\alpha \sim U(-\eta, 1 + \eta)$ for $\eta > 0$. For latent spaces whose dimension $m$ is large, in order to avoid the expensive and memory-consuming computation of $G(z) \in \mathbb{R}^{m \times m}$, we use the following regularization term in the subsequent experiments:

$$\mathbb{E}_{z \sim P}[\mathbb{E}_{v \sim \mathcal{N}(0,I)}[\|v^T G(z)v - cv^T v\|^2]], \qquad (10)$$

where we use mini-batch sampling to estimate the expectations. This can be done much more efficiently since we need only compute the Jacobian-vector product, i.e., $\sum_a (J_f)_a^{ij} v^a$.

## 5 EXPERIMENTAL RESULTS

We now verify the effectiveness of the info-Riemannian metric for the two point cloud autoencoder applications described above using both a synthetic 3D basic shape dataset and standard benchmark point cloud datasets.

The synthetic 3D basic shape dataset consists of cylinders, cubes, cones, and ellipsoids with various aspect ratios of the shape parameters (e.g., radius versus height for the cylinder). We sample 512 points from the surface mesh of the shapes using a greedy sample elimination algorithm, so that each sampled point is approximately the same distance from its neighborhood points (Yuksel, 2015). Each point cloud is then normalized so that distance between the two farthest points is one. Further details about the synthetic 3D basic shape dataset generation are provided in Appendix D.

The standard benchmark point cloud dataset consists of ModelNet (Wu et al., 2015) and ShapeNet (Chang et al., 2015), where ModelNet consists of ModelNet10 and ModelNet40, each of which consist of 10 and 40 shape classes, respectively. Each point cloud data has 2048 points; we normalize these into a unit sphere as done in (Yang et al., 2018b).

### 5.1 SYNTHETIC 3D BASIC SHAPE DATASET

In Section 5.1.1, we use a dataset consisting of cones, cylinders, and ellipsoids, which are split into training/validation/test sets of size 3196/800/804. We then confirm the validity of the proposed metric by comparing the results of several shape interpolation methods in the latent space.

In Section 5.1.2, to study the effects of the regularization term when learning the optimal latent coordinates (with respect to the info-Riemannian metric), we use a dataset consisting of boxes, cones, and ellipsoids divided into training/validation/test sets of size 720/240/240.

We use DGCNN as the encoder (Wang et al., 2019) and a fully-connected neural network as the decoder. The latent space is assumed to be two-dimensional. For the reconstruction loss term, we use the Chamfer distance. The regularization term in Equation (9) is multiplied by a coefficient $\lambda > 0$ and added to the reconstruction loss term. Further details about the network architectures and training are provided in Appendix D.

### 5.1.1 EXAMPLE 1: CONE, CYLINDER, AND ELLIPSOID

Figure 5 shows the test data encoded in the latent space together with the interpolation trajectories and generated point clouds from those interpolants. In the case of intra-class interpolations (i.e.,

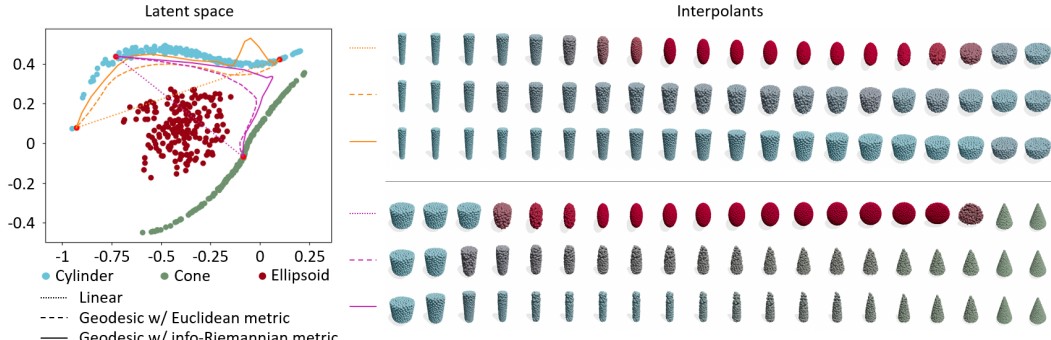

Figure 5: *Left*: Latent space with linear and geodesic interpolants. The orange interpolants connect a wide cylinder to a tall cylinder, while the magenta interpolants connect a cylinder to a cone. Linear interpolants and geodesic interpolants under the Euclidean and info-Riemannian metrics are drawn as dotted, dashed, and solid lines, respectively. *Right*: Generated point clouds from those interpolants. To visually indicate which class generated point cloud belong to, we color these according to the ratio of the Chamfer distances to the nearest point cloud for each class (see Appendix D). For example, when it is uncertain which class a generated data belongs to (i.e., the nearest distances to each class are similar), it is assigned some color other than blue, red, or green.

interpolants between cylinders), the linear interpolant clearly passes through the red ellipsoid region in the latent space, with some of the generated point clouds clearly ellipsoids. The geodesic interpolations under the Euclidean and info-Riemannian metrics both avoid ellipsoid regions in the latent space. In particular, the geodesic interpolants between two cylinders are also cylinders; this is well-aligned with human intuition. However, if we look at the generated point clouds in detail, while the info-Riemannian metric produces clearly blue cylinders, some of the generated point clouds with the Euclidean metric are non-blue cylinders (i.e., relatively closer to the ellipsoid region) with noisy side surfaces. For the inter-class interpolations (i.e., interpolants between a cylinder and a cone), the linear interpolant also clearly passes through the red ellipsoid region. The geodesic interpolation under the Euclidean metric produces many non-blue and non-green color shapes during the transition from cylinders to cones, while geodesic interpolation under the info-Riemannian metric produces such cases far less. Overall, it can be observed that the geodesic interpolants under the info-Riemannian metric have minimal shape class changes.

Figure 6 shows the latent spaces produced by the regularized autoencoders using the Euclidean and Info-Riemannian metrics. Compared to the latent space in Figure 5 (trained without regularization), the following observations can be made: (i) the encoded latent spaces of cones and cylinders are flattened, and (ii) the ellipsoids and cones become more discriminative. Furthermore, we note that the encoded latent space curves of cones and cylinders in the right figure (where the info-Riemannian metric is used) are clearly flatter than those in the left figure (where the Euclidean metric is used). In other words, if we linearly interpolate between

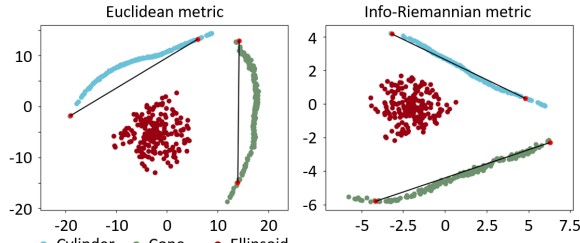

Figure 6: Latent spaces produced by regularized autoencoders, each of which is trained with the Euclidean (*Left*) and info-Riemannian metric (*Right*). Representative intra-class linear interpolants between two cylinders and two cones are drawn as black solid lines.

two cylinders or two cones, the interpolants in the right case will most likely remain in the same class, unlike the left case (the generated point clouds from the linear interpolants are provided in Appendix E).

### 5.1.2 EXAMPLE 2: BOX, CONE, AND ELLIPSOID

Figure 7 shows the test data encoded in the latent space together with the visualization of the Riemannian metric, the fitted Gaussian Mixture Model and its samples, and pairwise Euclidean dis-

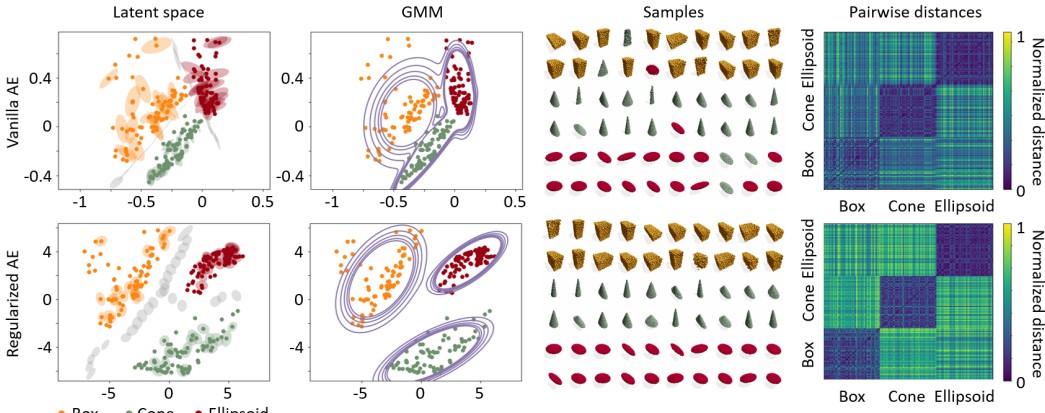

Figure 7: From *left* to *right*: latent spaces with equidistant ellipse ($\{z|(z-z^*)^T G(z^*)(z-z^*) = 1\}$ for center $z^*$) centered on some selected points and sampled points from interspaces, Gaussian Mixture Model (GMM) fitting results, generated samples from the GMM, and the heat map of the pairwise Euclidean distances in the latent space of all test data. The *upper* figure is a vanilla autoencoder trained without regularization, while the *lower* figure is trained with regularization (using the info-Riemannian metric). For the samples in the third column, we assign colors using the same method of Section 5.1.1 to visually express which classes the samples are likely to belong to.

tances. From the first column of Figure 7, by comparing the results with and without regularization, the following two key observations can be made: (i) in the vanilla autoencoder case (upper case), the major axes of the gray ellipses are aligned with the decision boundary (i.e., a hypersurface that partitions the different class regions), which implies that shapes of different classes are actually more distant in the learned manifold (under the info-Riemannian metric) than shown in the latent space, and (ii) in the regularized autoencoder case (lower case), by encouraging the metric to be isotropic (i.e., turning ellipses into circles), the gaps between different class regions are widened. The second column confirms that the components of the GMM are better separated after regularization; each component of the GMM on the regularized autoencoder generates high-quality, even samples from the same class shape as shown in the third column. The heat maps of the pairwise distances in the last column also indicate that shapes in different classes are more distant, and therefore more easily separable in the latent space of the regularized autoencoder.

## 5.2 STANDARD BENCHMARK DATA

To show that the regularization technique with the info-Riemannian metric can benefit unsupervised representation learning from the perspective of discriminative representation learning, we compare the transfer classification accuracy of ShapeNetCore.v2 to ModelNet following the same experimental procedure outlined in (Yang et al., 2018b). When training autoencoders with ShapeNet, random rotations about an axis parallel to the direction of gravity are applied to each point cloud. We use four different point cloud autoencoders: *FcNet* and *FoldingNet* adopted from (Yang et al., 2018b), *PointCapsNet* adopted from (Zhao et al., 2019), and *DGCNN-FcNet* using DGCNN (Wang et al., 2019); the latent space is 512-dimensional for all. The four autoencoders are trained with and without regularization. In the former case, the regularization terms of Equation (10) under both the Euclidean and info-Riemannian metrics are used while varying the regularization coefficient $\lambda$. We distinguish between regularized autoencoders using the Euclidean and info-Riemannian metrics by an "+E" or "+I" after the network name. After network training is finished, we train linear SVM classifiers with the encoded data for ModelNet10 and ModelNet40. These are split into training/test sets of sizes 3991/909 and 9843/2468, respectively. Further experimental detail are provided in Appendix D.

Table 1 shows a comparison of transfer classification accuracy from ShapeNet to ModelNet10 (MN10) and ModelNet40 (MN40) for various recent state-of-the-art methods. In the left column (*Adopted from References*), the numbers are adopted from previous papers (the experimental procedures may differ slightly from ours). In the right column (*Implemented by Authors*), we report the best numbers obtained (adopted from Appendix E). As shown in the right column, regularization using the info-Riemannian metric improves classification accuracy over vanilla autoencoders, with

Table 1: Classification accuracy by transfer learning for ModelNet10 (MN10) and ModelNet40 (MN40) from ShapeNet.

| METHOD | MN40 | MN10 | METHOD | MN40 | MN10 |
|---|---|---|---|---|---|
| *Adopted from References* | | | *Implemented by Authors* | | |
| SPH (Kazhdan et al., 2003) | 68.2% | 79.8% | FcNet | 88.3% | 93.5% |
| LFD (Chen et al., 2003) | 75.5% | 79.9% | FcNet + E (ours) | 89.3% | 93.7% |
| VConv-DAE (Sharma et al., 2016) | 75.5% | 80.5% | FcNet + I (ours) | 90.4% | 94.3% |
| 3D-GAN (Wu et al., 2016) | 83.3% | 91.0% | FoldingNet | 89.3% | 93.7% |
| Latent-GAN (Achlioptas et al., 2018) | 84.5% | **95.4%** | FoldingNet + E (ours) | 88.9% | 94.4% |
| FoldingNet (Yang et al., 2018b) | 88.4% | 94.4% | FoldingNet + I (ours) | 90.1% | 94.5% |
| PointFlow (Yang et al., 2019) | 86.8% | 93.7% | PointCapsNet | 87.2% | 93.6% |
| Multi-Task (Hassani & Haley, 2019) | 89.1% | - | PointCapsNet + E (ours) | 88.1% | 93.7% |
| PointCapsNet (Zhao et al., 2019) | **89.3%** | - | PointCapsNet + I (ours) | 88.5% | 93.9% |
| | | | DGCNN-FcNet | 90.3% | 94.5% |
| | | | DGCNN-FcNet + E (ours) | 89.9% | 94.4% |
| | | | DGCNN-FcNet + I (ours) | **91.0%** | **95.2%** |

higher accuracy compared to the Euclidean metric case. Although their performance is not directly comparable due to differences in the experimental procedures, it can be seen that our regularized autoencoders are comparable to the state-of-the-art methods and, at least for our implementation, shows higher classification accuracy than vanilla autoencoders.

We also conduct additional experiments to determine how much more robust the representation obtained with our regularization approach is for noisy point cloud data. We add noise with different levels of standard deviation (1%, 5%, 10%, and 20% of the diagonal length of the point cloud bounding box) to point cloud data (see Appendix D for details). Then *FcNet* is trained with and without regularization in the same way as above.

Table 2: Classification accuracy by transfer learning for ModelNet10 (MN10) and ModelNet40 (MN40) from ShapeNet under the noise levels of 1%, 5%, 10%, and 20%.

| METHOD | MN40 | | | | MN10 | | | |
|---|---|---|---|---|---|---|---|---|
| | 1% | 5% | 10% | 20% | 1% | 5% | 10% | 20% |
| FcNet | 87.8% | 83.2% | 75.6% | 64.5% | 92.4% | 91.9% | 88.4% | 79.8% |
| FcNet + E (ours) | 86.6% | 85.1% | 79.1% | 70.4% | 92.2% | 91.1% | 88.2% | 82.6% |
| FcNet + I (ours) | **89.0%** | **86.6%** | **81.4%** | **72.4%** | **93.3%** | **92.6%** | **91.6%** | **84.8%** |

Table 2 shows a comparison of transfer classification accuracy in the presence of noise. As the noise level increases, the classification accuracy obviously decreases in both cases, but the reduction is more dramatic for the vanilla autoencoder case. Additional experimental results including semi-supervised classification task are included in Appendix E. Overall, it is indeed somewhat surprising that unsupervised classification accuracy can be improved with a simple regularization technique in lieu of a complex neural network architecture or loss function.

## 6 DISCUSSION AND CONCLUSION

This paper has proposed a new Riemannian geometric structure for the space of point cloud data. We have defined a statistical representation of point cloud data and constructed a statistical manifold in a mathematically rigorous way. Then a natural Riemannian metric – Fisher information metric – is assigned to the point cloud statistical manifold, which provides geometrically well-defined measures needed for applications. We demonstrate its advantages through two applications involving point cloud autoencoders: (i) minimal geodesic interpolants under info-Riemannian metric have minimal shape changes compared to the standard linear interpolants, and (ii) the optimal latent coordinates learned using our method produce more discriminative representation spaces than existing methods.

As a potential issue, the "fixed number of points" assumption used in our construction of the statistical manifold may be violated in real world problems. In such cases, we can easily mitigate this issue by matching the number of points in each point cloud through a simple upsampling/down-sampling algorithm. Further, the kernel function used in our current implementations, the standard normal kernel function, may not be an optimal choice. Other choices can be explored to enhance our algorithms as long as the conditions of Proposition 2 are satisfied.

REPRODUCIBILITY STATEMENT

We have provided clear explanations of the assumptions and statements for the theoretical results, and included complete proofs of the propositions in Appendix B. Also, we have included experiment settings in detail as much as possible in Appendix C and D such as the number of training/validation/test splits of datasets, data pre-processing methods, neural network architectures, and hyperparameters used in training (e.g., batch size, number of epochs, learning rate, etc). In particular, we have included the details of the synthetic dataset generation in Appendix D.

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

APPENDIX

## A EXISTING GEOMETRIC/STATISTICAL METHODS FOR POINT CLOUD DATA

### A.1 GEOMETRIC METHODS

The Hausdorff distance measures the distance between two non-empty subsets of a metric space (Hausdorff, 1914; 2008). Given two point clouds $\mathbf{X} = \{x_1, ..., x_n \mid x_i \in \mathbb{R}^D\}$ and $\mathbf{Y} = \{y_1, ..., y_n \mid y_i \in \mathbb{R}^D\}$ and metric $\|x - y\|^2$ in $\mathbb{R}^D$, the Hausdorff distance can be computed as follows:

$$\max(\max_{x \in \mathbf{X}}(\min_{y \in \mathbf{Y}} \|x - y\|^2), \max_{y \in \mathbf{Y}}(\min_{x \in \mathbf{X}} \|x - y\|^2)).$$

The Hausdorff distance is susceptible to outliers; hence, in practice, the average Hausdorff distance is used more often:

$$\frac{1}{|\mathbf{X}|} \sum_{x \in \mathbf{X}} \min_{y \in \mathbf{Y}} \|x - y\|^2 + \frac{1}{|\mathbf{Y}|} \sum_{y \in \mathbf{Y}} \min_{x \in \mathbf{X}} \|x - y\|^2,$$

where $|\mathbf{X}|$ denotes the number of elements in the set $\mathbf{X}$. A slightly modified version of this is often referred to as the Chamfer distance (Yang et al., 2018b). The popular point cloud registration algorithm ICP relies on these classes of metrics (Besl & McKay, 1992).

Another popular similarity measure between two point cloud data is the Earth Mover's Distance (EMD) (Rubner et al., 2000):

$$\sum_{x \in \mathbf{X}} \min_{\phi:\mathbf{X} \to \mathbf{Y}} \|x - \phi(x)\|^2,$$

where $\phi$ is a bijective mapping. Although the EMD is computationally more expensive than the above Hausdorff distances, comparing point clouds with optimal matching in EMD provides a more robust and well-behaved similarity measure.

The Chamfer distance and EMD are often used to measure the distances between two point clouds, but they are typically computationally expensive. Recently, the sliced Wasserstein distance and its variants have been proposed to more efficiently measure distances (Nguyen et al., 2021). In another study, each point cloud data is represented as a matrix of pairwise Euclidean distances between all points, and the Frobenius norm of the difference between the two matrices is used as the distance between two point clouds Cosmo et al. (2020).

However, all these distance metrics measure just one aspect of point cloud data; other applications may require more advanced concepts and tools. For example, in the case of a moving point cloud, one may wish to measure some of its more dynamic aspects like its velocity or other quantities that require a notion of higher-order derivatives. Applications such as Monte Carlo sampling may require the construction of an isotropic Gaussian distribution on the underlying space, in which case the notion of "angle" is needed in addition to distance (more technically, an inner product on the tangent space is required (Girolami & Calderhead, 2011; Gemici et al., 2016; Mallasto & Feragen, 2018).

We believe a rigorous mathematical characterization of the space of point cloud data is an essential ingredient to a more comprehensive, correct, and robust analysis of the types described above, particularly one based on Riemannian geometry. In this paper, we rigorously define a Riemannian geometric structure for the space of point cloud data.

### A.2 STATISTICAL METHODS

Interpreting the point cloud data as a set of samples from some underlying probability distribution is very intuitive and natural, and has been adopted in many previous works (Jian & Vemuri, 2005; Wang et al., 2006; Myronenko & Song, 2010; Hasanbelliu et al., 2014; Zhou et al., 2014; Min et al., 2018; Li et al., 2021). Commonly, a mixture model is used to describe the point cloud, written as follows:

$$p(x; w, \theta) := \sum_{i=1}^{k} w_i \phi(x|\theta_i),$$

where the $w_i$ are weights and $\phi(x|\theta_i)$ are primitive density functions with parameter $\theta_i$ (e.g., $\phi(x|\theta_i)$ can be a standard Gaussian with mean $\theta_i$). Given a point cloud $\mathbf{X} := \{x_1, \dots, x_n | x_i \in \mathbb{R}^D\}$, the parameters $w, \theta$ are either fit with data or specified by the user, then the mixture model is used as a statistical representation of the point cloud data. Besides the most popular choice for $\phi$, the Gaussian (Jian & Vemuri, 2005), other choices such as the t-distribution (Zhou et al., 2014) or hybrid model (Min et al., 2018) have been explored.

The main purpose behind using statistical representations in existing works are to use the well-known information-theoretic divergence measures such as the KL-divergence to compute the similarity between point clouds. However, these divergence measures (which are approximate distance metrics) only measure just one aspect of point cloud just like the distance measures discussed in A.1. Effective mathematical concepts and tools for defining and measuring other important geometric aspects of point cloud are still lacking.

# B  PROOF OF THE PROPOSITIONS

## B.1  PROOF OF PROPOSITION 2

*Proof.* Let's consider two point clouds $\{y_i\}_{i=1}^n$ and $\{z_i\}_{i=1}^n$. To show that the mapping $h : \mathcal{X} \to \mathcal{S}$ is 1-1 (especially injective), we have to prove the following statement:

$$p(x; \{y_i\}_{i=1}^n) = p(x; \{z_i\}_{i=1}^n) \implies \{y_i\}_{i=1}^n = \{z_i\}_{i=1}^n. \tag{11}$$

The conditional statement can be rewritten as follows:

$$\sum_{i=1}^n K(\Sigma^{-\frac{1}{2}}(x - y_i)) = \sum_{i=1}^n K(\Sigma^{-\frac{1}{2}}(x - z_i)). \tag{12}$$

Let denote $B = \{y_i\}_{i=1}^n \cap \{z_i\}_{i=1}^n$ and $|B| = m$, and assume that $m < n$. Then the above equation is reduced to

$$\sum_{y \in \{y_i\}_{i=1}^n - B} K(\Sigma^{-\frac{1}{2}}(x - y)) - \sum_{z \in \{z_i\}_{i=1}^n - B} K(\Sigma^{-\frac{1}{2}}(x - z)) = 0. \tag{13}$$

Since the sets $\{y_i\}_{i=1}^n - B$ and $\{z_i\}_{i=1}^n - B$ are disjoint and each set has $n - m$ elements, the LHS has $2(n - m) \leq 2n$ terms and the terms are different to each other. Then, by the assumption, the above $2(n - m)$ terms are linearly independent, so the above equation cannot hold by the definition of the linear independence. In the other words, there is a contradiction and we can find that the assumption $m < n$ is wrong. Therefore, $m = n$, so $\{y_i\}_{i=1}^n = \{z_i\}_{i=1}^n$. $\qquad \square$

## B.2  PROOF OF PROPOSITION 3

*Proof.* First, let's think about the 1-dimensional case, i.e., $x \in \mathbb{R}$. Consider an arbitrary point set $\{x_i\}_{i=1}^N$ and the corresponding set of functions $\{K(\sigma^{-\frac{1}{2}}(x - x_i))|i = 1, 2, ..., N\}$, where the kernel function $K$ is

$$K(u) = \frac{1}{\sqrt{2\pi}} \exp(-\frac{u^2}{2}). \tag{14}$$

Suppose there exist non-zero constants $a_1, a_2, ..., a_N$ such that

$$\sum_{i=1}^N a_i K(\sigma^{-\frac{1}{2}}(x - x_i)) = \sum_{i=1}^N \frac{1}{\sqrt{2\pi\sigma}} a_i \exp(-\frac{(x - x_i)^2}{2}) = 0 \ \forall x \in \mathbb{R}. \tag{15}$$

The above expression can be reduced as follows:

$$\exp(-\frac{x^2}{2}) \sum_{i=1}^N a_i \exp(x_i x) \exp(-\frac{x_i^2}{2}) = 0 \ \forall x \in \mathbb{R}. \tag{16}$$

Let $\lambda_i = a_i \exp(-\frac{x_i^2}{2})$ for $i = 1, 2, ..., N$, then the above equation is simplified as:

$$\sum_{i=1}^N \lambda_i \exp(x_i x) = 0 \ \forall x \in \mathbb{R}. \tag{17}$$

In the other words, it is enough to show that the functions in the set $\{\exp(x_i x)|i = 1, 2, ..., N\}$ are linearly independent.

There is a useful tool for proving the linear independence of the functions:

**Theorem 1** (Wronskian's Theorem). *Let $f_1, f_2, ..., f_n : I \to \mathbb{R}$ be $n - 1$ times differentiable functions on an interval $I$ and the Wronskian $W(f_1, f_2, ..., f_n)$ be a function on I defined by:*

$$W(f_1, f_2, ..., f_n)(x) = \begin{vmatrix} f_1(x) & f_2(x) & \ldots & f_n(x) \\ f_1'(x) & f_2'(x) & \ldots & f_n'(x) \\ f_1''(x) & f_2''(x) & \ldots & f_n''(x) \\ \vdots & \vdots & \ddots & \vdots \\ f_1^{(n-1)}(x) & \ldots & \ldots & f_n^{(n-1)}(x) \end{vmatrix} \quad \forall x \in \mathbb{R}, \tag{18}$$

*where this is the determinant of the square matrix constructed by placing the functions and their derivatives in an appropriate way. Then, if the Wronskian of this set of functions is not identically zero, then the set of functions is linearly independent.*

Using the above theorem, we can construct the Wronskian $W(x)$ with the set of functions $\{\exp(x_i x)|i = 1, 2, ..., N\}$. Substituting zero for $x$, then we can obtain:

$$W(0) = \begin{vmatrix} 1 & 1 & \ldots & 1 \\ x_1 & x_2 & \ldots & x_N \\ x_1^2 & x_2^2 & \ldots & x_N^2 \\ \vdots & \vdots & \ddots & \vdots \\ x_1^{N-1} & \ldots & \ldots & x_N^{N-1} \end{vmatrix} \tag{19}$$

The above can be simplified by the mathematical induction:

$$\begin{vmatrix} 1 & 1 & \ldots & 1 \\ x_1 & x_2 & \ldots & x_N \\ x_1^2 & x_2^2 & \ldots & x_N^2 \\ \vdots & \vdots & \ddots & \vdots \\ x_1^{N-1} & \ldots & \ldots & x_N^{N-1} \end{vmatrix} = \begin{vmatrix} 1 & 0 & \ldots & 0 \\ x_1 & x_2 - x_1 & \ldots & x_N - x_1 \\ x_1^2 & x_2^2 - x_1^2 & \ldots & x_N^2 - x_1^2 \\ \vdots & \vdots & \ddots & \vdots \\ x_1^{N-1} & \ldots & \ldots & x_N^{N-1} - x_1^{N-1} \end{vmatrix} \tag{20}$$

$$= \begin{vmatrix} 1 & 0 & \ldots & 0 \\ x_1 & x_2 - x_1 & \ldots & x_N - x_1 \\ x_1^2 & x_2^2 - x_1^2 & \ldots & x_N^2 - x_1^2 \\ \vdots & \vdots & \ddots & \vdots \\ x_1^{N-1} & \ldots & \ldots & x_N^{N-1} - x_1^{N-1} \end{vmatrix} \tag{21}$$

$$= (x_2 - x_1) \cdots (x_n - x_1) \begin{vmatrix} 1 & \ldots & 1 \\ x_2 & \ldots & x_N \\ x_2^2 & \ldots & x_N^2 \\ \vdots & \ddots & \vdots \\ x_2^{N-2} & \ldots & x_N^{N-2} \end{vmatrix} \tag{22}$$

$$= \cdots \tag{23}$$

$$= \prod_{\substack{i,j=1 \\ i>j}}^{n} (x_i - x_j) \tag{24}$$

Since all the points $\{x_i\}_{i=1}^N$ are distinct to each other, the above determinant value must be non-zero. Then by Wronskian's Theorem, the functions $\{\exp(x_i x)|i = 1, 2, ..., N\}$ are linearly independent, so are the functions $\{K(\sigma^{-\frac{1}{2}}(x - x_i))|i = 1, 2, ..., N\}$.

Next, let's think about the general $D$-dimensional case with $D \geq 2$; $x \in \mathbb{R}^D$. The Wronskian's Theorem introduced above cannot be used, instead, we adopt a generalized Wronskian's Theorem applicable to multivariable functions.

**Theorem 2** (Generalized Wronskian's Theorem). *Let $f_1, f_2, ..., f_n : \mathbb{R}^D \to \mathbb{R}$ be multivariable functions and the generalized Wronskian $gW(f_1, f_2, ..., f_n)$ as a function defined by:*

$$gW(f_1, f_2, ..., f_n)(x) = \begin{vmatrix} f(x) \\ D_1 f(x) \\ D_2 f(x) \\ \vdots \\ D_{n-1} f(x) \end{vmatrix} \quad \forall x \in \mathbb{R}^D, \tag{25}$$

*where $f(x) = (f_1(x), ..., f_n(x))$, $D_j f(x)$ are row vectors, $D_j$ is any partial derivative of order not greater than $j$, and all $D_j$ are distinct for $j = 1, 2, ..., n-1$. We note that if at least one of the generalized Wronskians of this set of functions is not identically zero, then the set of functions is linearly independent.*

Similar to the 1-dimensional case $x_i \in \mathbb{R}$, consider an arbitrary point set $\{x_i\}_{i=1}^N$ and the corresponding set of functions $\{K(\sigma^{-\frac{1}{2}}(x - x_i))|i = 1, 2, ..., N\}$, where the kernel function $K$ is now:

$$K(u) = \frac{1}{\sqrt{(2\pi)^D}} \exp(-\frac{u^T u}{2}). \tag{26}$$

Suppose that there exist non-zero constants $b_1, b_2, ..., b_N$ such that

$$\sum_{i=1}^N b_i K(\sigma^{-\frac{D}{2}}(x - x_i)) = \sum_{i=1}^N \frac{1}{\sqrt{(2\pi\sigma)^D}} b_i \exp(-\frac{(x-x_i)^T(x-x_i)}{2}) = 0 \ \forall x \in \mathbb{R}^D. \tag{27}$$

The above expression can be reduced as follows:

$$\exp(-\frac{x^T x}{2}) \sum_{i=1}^N b_i \exp(x_i^T x) \exp(-\frac{x_i^T x_i}{2}) = 0 \ \forall x \in \mathbb{R}^D \tag{28}$$

Let $\omega_i = b_i \exp(-\frac{x_i^T x_i}{2})$ for $i = 1, 2, ..., N$. Then the final expression of the equation is as follows:

$$\sum_{i=1}^N \omega_i \exp(x_i^T x) = 0 \ \forall x \in \mathbb{R}^D. \tag{29}$$

In other words, it is enough to show that the functions $\{\exp(x_i^T x)|i = 1, 2, ..., N\}$ are linearly independent.

To apply the generalized Wronskian theorem, we need to find a proper set of partial derivatives. To find such set, our strategy is to use a coordinate transformation with a rotation matrix $R \in \mathrm{SO}(D)$, i.e., rotating vectors. To be more specific, we will find $R$ that makes the first components of $y_i = R x_i$ for $i = 1, 2, ..., N$ all different and use the transformed coordinates in the following proof. For this purpose, we first show that such rotation matrix $R$ exists.

Intuitively, we can find a vector $r_1$ so that $x_1, ..., x_n$ are projected to $r_1$ as all distinct points. If $R$ has this $r_1$ vector as the first column vector, then $R$ can make the first components of $y_i = R x_i$ for $i = 1, 2, ..., N$ all different. More precisely, the first components of the two points $x_i, x_j$ are same if $r_1$ is in the $(D-1)$-dimensional hyperplane $\{r_1 \in \mathbb{R}^D | r_1^T(x_i - x_j) = 0\}$. Since there are only $N(N-1)/2$ combinations of the pairs of $(x_i, x_j)$ for $i, j = 1, 2, ..., n$, we can find a vector $r_1$ such that

$$r_1^T(x_i - x_j) \neq 0 \ \ \forall i \neq j \ \text{ and } \ i, j = 1, 2, ..., n. \tag{30}$$

We pick a coordinate transform matrix $R \in \mathrm{SO}(D)$ with this $r_1$ as the first column vector. Then this $R$ can make the first components of $y_i = R x_i$ for $i = 1, 2, ..., N$ all different.

Using the $R$, rewrite the set of functions $\{\exp(x_i^T x)|i = 1, 2, ..., N\}$ as follows:

$$\{\exp(x_i^T x)|i = 1, 2, ..., N\} = \{\exp(x_i^T R^T R x)|i = 1, 2, ..., N\} \tag{31}$$

$$= \{\exp(y_i^T z)|i = 1, 2, ..., N\}, \tag{32}$$

where $z = Rx$ and $y_i = Rx_i$ for $i = 1, 2, ..., N$. We again note that the first elements of $y_i$ for $i = 1, ..., N$ are all distinct to each other.

To use the Theorem 2 in the transformed coordinates $z$, we can pick a generalized Wronskian $gW(z)$ with the corresponding partial derivatives:

$$D_1 = \frac{\partial}{\partial z_1}, D_2 = \frac{\partial^2}{\partial z_1^2}, ..., D_{n-1} = \frac{\partial^{n-1}}{\partial z_1^{n-1}}, \tag{33}$$

where $z = (z_1, ..., z_D)$. With this generalized Wronskian and the set of functions $\{\exp(y_i^T z)|i = 1, 2, ..., N\}$, substituting zero vector for $z \in \mathbb{R}^D$, we can obtain:

$$gW(0) = \begin{vmatrix} 1 & 1 & \cdots & 1 \\ y_{11} & y_{12} & \cdots & y_{1N} \\ y_{11}^2 & y_{12}^2 & \cdots & y_{1N}^2 \\ \vdots & \vdots & \ddots & \vdots \\ y_{11}^{N-1} & \cdots & \cdots & y_{1N}^{N-1} \end{vmatrix}, \tag{34}$$

where $y_{1i}$ is the first element of $y_i$ for $i = 1, 2, ..., N$. Since $y_{1i}$'s are all distinct to each other, we can obtain, just like the 1-dimensional case:

$$gW(0) = \prod_{\substack{i,j=1 \\ i>j}}^{n} (y_{1i} - y_{1j}) \neq 0. \tag{35}$$

By generalized Wronskian's Theorem, the functions $\{\exp(x_i^T x)|i = 1, 2, ..., N\}$ are linearly independent, so are the functions $\{K(\sigma^{-\frac{1}{2}}(x - x_i))|i = 1, 2, ..., N\}$, also in the $D$-dimensional case. $\qquad\square$

### B.3   PROOF OF PROPOSITION 4

*Proof.* Since $\frac{\partial \log p(x;X)}{\partial X^{ij}} = \frac{1}{p(x;X)} \frac{\partial p(x;X)}{\partial X^{ij}}$, the Riemannian metric $H_{ijkl}$ in equation (4) is

$$\int p(x;X) \frac{1}{p^2(x;X)} \frac{\partial p(x;X)}{\partial X^{ij}} \frac{\partial p(x;X)}{\partial X^{kl}} \, dx.$$

By plugging $p(x;X)$ in equation (3) in $\frac{\partial p(x;X))}{\partial X^{ij}}$, we get the following expression:

$$\begin{aligned} n\sqrt{|\Sigma|} \frac{\partial p(x;X)}{\partial X^{ij}} &= \frac{\partial}{\partial X^{ij}} \sum_{a=1}^{n} K(\Sigma^{-1/2}(x - x_a)) \\ &= \frac{\partial}{\partial X^{ij}} K(\Sigma^{-1/2}(x - x_i)) \\ &= J_K(\Sigma^{-1/2}(x - x_i))\Sigma^{-1/2} \frac{\partial}{\partial X^{ij}}(x - x_i) \\ &= J_K(\Sigma^{-1/2}(x - x_i))\Sigma^{-1/2} \frac{\partial}{\partial X^{ij}}(x - x_i) \\ &= -[J_K(\Sigma^{-1/2}(x - x_i))\Sigma^{-1/2}]_j, \end{aligned}$$

$J_K : \mathbb{R}^D \to \mathbb{R}^D$ is the Jacobian of the kernel function $K$. This consequently leads to

$$H_{ijkl}(X) := \int p(x;X) \frac{(J_K|_{h(x,x_i)}\Sigma^{-\frac{1}{2}})_j (J_K|_{h(x,x_k)}\Sigma^{-\frac{1}{2}})_l}{(\sum_{m=1}^{n} K(h(x,x_m)))^2} \, dx,$$

where $h(x, x_i) = \Sigma^{-\frac{1}{2}}(x - x_i)$.

If $K$ is the standard normal kernel function, then we get $J_K(x) = -K(x)x^T$. By plugging this in the above equation with $\Sigma = \sigma^2 I$, we get the following:

$$
\begin{aligned}
H_{ijkl}(X) &= \int p(x; X) \frac{(J_K|_{h(x,x_i)} \Sigma^{-\frac{1}{2}})_j (J_K|_{h(x,x_k)} \Sigma^{-\frac{1}{2}})_l}{(\sum_{m=1}^n K(h(x, x_m)))^2} \, dx \\
&= \int p(x; X) \frac{K(h(x, x_i)) K(h(x, x_k))}{(\sum_{m=1}^n K(h(x, x_m)))^2} \left[ \frac{(x - x_i)(x - x_k)^T}{\sigma^4} \right]_{jl} \, dx.
\end{aligned}
$$

$\square$

## C  RANDOM WALKS OVER THE POINT CLOUD RIEMANNIAN MANIFOLD

This section describes details of the random walk experiments over the point cloud statistical manifold equipped with a Riemannian metric $H_{ijkl}, i, k = 1, ..., n, j, l = 1, 2, 3$. First, an initial shape is defined by slightly deforming the reference sphere with the addition of a small velocity matrix $V$. Then, sequences of point clouds are generated by repeating the following process: i) given a point cloud matrix $X$, reorder it as a $3n$-dimensional vector and the Riemannian metric at $X$ as $3n \times 3n$ matrix, ii) sample a velocity vector from the Gaussian distribution whose covariance is the inverse of the Riemannian metric, iii) reorder the sampled velocity vector to $n \times 3$ matrix representation and divide it by its norm induced by the metric, and iv) add it to $X$.

## D  IMPLEMENTATION DETAILS FOR THE EXPERIMENTS

### D.1  SYNTHETIC 3D BASIC SHAPE DATASET GENERATION

For the synthetic 3D basic shape dataset, we define 5 shape classes consist of cylinder, cone, elliptic cone, ellipsoid, and box. Figure 8 shows the representative shape of the shape corresponding to each class and the shape parameters required to define the shape class. We again note that we sample 512 points from the surface mesh of the shapes using a greedy sample elimination algorithm, and each point cloud is then normalized so that distance between the two farthest points is one. In the process of normalizing the point cloud, the dimension of the shape parameter is reduced by one dimension. As an example, in the case of cylinder, the cylinders are contained in a two-dimensional space defined by a radius dimension $r$ and a height dimension $h$, but it is reduced to one dimension through normalization. In summary, cylinders and cones with two shape parameters are placed in one-dimensional space, and elliptic cones, ellipsoids, and boxes with three parameters are placed in two-dimensional space.

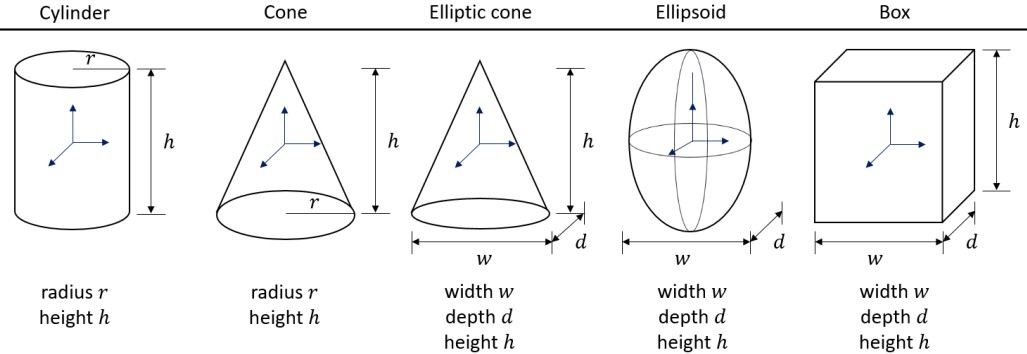

Figure 8: Representative shape to each class and the shape parameters required to define it. There are 5 shape classes including cylinder, cone, elliptic cone, ellipsoid, and box.

In Section 5.1.1, we use a dataset consisting of cones, cylinders, and ellipsoids, which are split into training/validation/test sets of size 3196/800/804. The detail ranges of the exact value of the shape parameters are shown in Table 3.

Table 3: The ranges of the shape parameters of the dataset used in Section 5.1.1

| SHAPE | param | min | max | param | min | max | param | min | max |
|---|---|---|---|---|---|---|---|---|---|
| Cylinder | $r$ | 0.01 | 0.12 | $h$ | 0.05 | 0.45 | | | |
| Cone | $r$ | 0.02 | 0.15 | $h$ | 0.02 | 0.45 | | | |
| Ellipsoid | $w$ | 0.03 | 0.12 | $d$ | 0.03 | 0.12 | $h$ | 0.03 | 0.12 |

In Section 5.1.2, we use a dataset consisting of boxes, cones, and ellipsoids divided into training/validation/test sets of size 720/240/240. The detail ranges of the aspect ratios of the shape parameters are shown in Table 4.

Table 4: The ranges of the shape parameters of the dataset used in Section 5.1.2

| SHAPE | param | min | max | param | min | max |
|---|---|---|---|---|---|---|
| Elliptic cone | $d/w$ | 0.33 | 3 | $h/w$ | 0.33 | 3 |
| Ellipsoid | $d/w$ | 0.33 | 3 | $h/w$ | 0.125 | 0.33 |
| Box | $d/w$ | 0.33 | 3 | $h/w$ | 0.33 | 3 |

## D.2 DETAILS FOR EXPERIMENTS ON SYNTHETIC 3D BASIC SHAPE DATASET

We used an encoder with a structure similar to the classification network used in DGCNN (Wang et al., 2019). The input point cloud with dimension 3×512 passes through five EdgeConv layers with point-wise latent space dimensions (64, 64, 128, 256) and a max pooling layer (we do not use a batch normalization layer unlike the original DGCNN classification network), then we can obtain a 1024-dimensional feature vector. Other settings are the same (e.g., $k = 20$, leaky relu activation) Then this feature vector again passes through three fully-connected neural networks with dimension (512, 256, 2) with leaky relu activation function and linear output activation function; the latent space is two-dimensional. For the decoder model, we simply use a fully-connected neural network as the decoder. The two-dimensional vector on the latent space passes through three fully-connected neural networks with dimension (256, 512, 3×512) with relu activation function and linear output activation function; the output is a 3D point cloud with the number of points 512.

**Section 5.1.1:** To train the networks, we use ADAM with a learning rate of 0.001 and batch size of 16; the total number of the epochs is 500. The mean value of MEDs of the dataset is 0.0339, and we use the bandwidth value $k$ to 0.5. We use Chamfer distance as the reconstruction loss; for regularization figures, the regularization term with the version of Equation (9) is multiplied by a coefficient $\lambda = 10^7$ with the info-Riemannian metric and by a coefficient $\lambda = 1$ with the Euclidean metric and added to the reconstruction loss term for each metric case. The value of $\eta$ is set to be 0.0. For geodesic computation, we parametrize the curve $z(t)$ by a cubic spline with fixed boundary points $z_1, z_2$ and 10 control points. The control points are first initialized with equally spaced linear interpolants between $z_1$ and $z_2$. Then, for each iteration of optimization, we randomly sample 40 points on $t_i \sim U(0, 1), i = 1, ..., 40$ and calculate an expectation $\frac{1}{40} \sum_{i=1}^{40} \dot{z}(t_i)^T G(z(t_i)) \dot{z}(t_i)$ over the sampled points as the approximation of the objective function. We use ADAM with a learning rate of 0.001 and the total number of the iterations is 5000.

**Section 5.1.2:** To train the networks, we use ADAM with a learning rate of 0.001 and batch size of 16; the total number of the epochs is 3000. The mean value of MEDs of the dataset is 0.0341, and we use the bandwidth value $k$ to 0.5. We use Chamfer distance as the reconstruction loss, and the regularization term with the version of Equation (9) is multiplied by a coefficient $\lambda = 10^7$ and added to the reconstruction loss term. The value of $\eta$ is set to be 0.0.

**Color assigning method:** To visually indicate which class generated point cloud belong to, we color these according to the ratio of the Chamfer distances to the nearest point cloud for each class. In detail, the smallest value (distance to nearest point cloud) is found by comparing the distance between the given point cloud and all point clouds of each class in the dataset. Since we are using 3 shape classes in both examples, we call the nearest distance to each class $d_1, d_2$, and $d_3$. After

that, the vector $d = (d_1, d_2, d_3)$ is normalized with 2-norm so that the 2-norm of the vector to be 1. Finally, the value $0.2 \times \text{Softmax}(1/d_1, 1/d_2, 1/d_3)$ is regarded as the ratio of the distances and a color is assigned to a given point cloud according to this ratio (i.e., linear weighted sum in the RGB coordinate).

### D.3 DETAILS FOR EXPERIMENTS ON STANDARD BENCHMARK DATASET

We use four different point cloud autoencoders: *FcNet*, *FoldingNet*, *PointCapsNet*, and *DGCNN-FcNet*; the latent space is 512-dimensional. For *FcNet* and *FoldingNet*, we use the exactly same point cloud autoencoder structures both adopted from (Yang et al., 2018b). For *PointCapsNet*, we also use the exactly same point cloud autoencoder structure adopted from (Zhao et al., 2019); we use $16 \times 32$ capsules to restrict the latent space to a reasonable size of 512. For *DGCNN-FcNet*, we use DGCNN classification network as encoder (i.e., the same encoder architecture used in experiments on synthetic 3D basic shape dataset, see Appendix C.2), and the same decoder structure from *FcNet* as decoder (i.e., three fully-connected neural networks with dimension (1024, 2048, 3×2048) with relu activation function and linear output activation function). To train the networks, we use ADAM with a learning rate of 0.0001, betas of $[0.9, 0.999]$, and weight decay of 0.000001 and batch size of 16; the total number of the epochs is 500. The mean value of MEDs of the dataset is 0.0356, and we use the bandwidth value $k$ to 0.8. We use Chamfer distance as the reconstruction loss and regularization term with the version of Equation (10) with the value of $\eta$ to be 0.2. The regularization term is multiplied by various coefficients, where the values of the regularization coefficients are summarized in Appendix D.

### D.4 DETAILS FOR EXPERIMENTS ON STANDARD BENCHMARK DATASET WITH NOISE

We use the exactly same point cloud autoencoder structures adopted from (Yang et al., 2018b), *FcNet*, where the latent space is 512-dimensional. We add noise to each point **x** in point cloud of the dataset (ShapeNet, ModelNet10, and ModelNet40) according to $\mathbf{x} \mapsto \mathbf{x} + m\mathbf{v}$, where **v** is uniformly sampled on the unit sphere and $m$ is sampled from the Gaussian distribution with zero mean and different levels of standard deviation (1%, 5%, 10%, and 20% of the diagonal length of the point cloud bounding box). The training configuration is the same with the case of Appendix D.3 except the followings. The regularization term is multiplied by $\lambda = 8000$. The mean values of MEDs of the dataset are 0.0320, 0.0364, 0.0442, and 0.579 for the cases of the noise levels 1%, 5%, 10%, and 20%, respectively, and we use the bandwidth value $k$ to 0.8.

## E ADDITIONAL EXPERIMENTAL RESULTS

### E.1 SYNTHETIC DATASET

#### E.1.1 QUANTITATIVE RESULTS

In this section, we quantitatively evaluate how much the regularization term improves class separability. In addition to the example shown in Figure 7, more diverse synthetic datasets are made and experiments are conducted. We use datasets consisting of boxes, elliptic cones, and ellipsoids as the case in Figure 7. But in these additional experiments, for each shape class, we use more diverse parameter configurations. In details, we generate short, normal, and tall shapes for each shape class, and the aspect ratios of the shape parameters are shown in Table 5. We conduct a total of 27 experiments with $3^3$ combinations.

Similar to the experiment of Section 5.1.2, each dataset is divided into training/validation/test sets of size 720/240/240. Training configurations are the same with the experiment of Section 5.1.2, except that the mean values of MEDs are different to each other (but we consistently use the bandwidth value $k$ to 0.5) and the total number of epochs is 500.

In the obtained representation spaces, after fitting the Gaussian mixture model by using the training and validation data, the clustering scores are measured with the test data. The Normalized Mutual Information (NMI), Adjusted Rand Index (ADI), and Silhouette Coefficient (SC) are used as metrics. The results are shown in Table 6. For all three clustering measures, regularized autoencoder shows

Table 5: The ranges of the shape parameters of the dataset used in quantitative analysis on synthetic dataset

| SHAPE | param | min | max | param | min | max |
|---|---|---|---|---|---|---|
| Elliptic cone short | $d/w$ | 0.33 | 3 | $h/w$ | 0.125 | 0.33 |
| Elliptic cone normal | $d/w$ | 0.33 | 3 | $h/w$ | 0.33 | 3 |
| Elliptic cone tall | $d/w$ | 0.33 | 3 | $h/w$ | 3 | 8 |
| Ellipsoid short | $d/w$ | 0.33 | 3 | $h/w$ | 0.125 | 0.33 |
| Ellipsoid normal | $d/w$ | 0.33 | 3 | $h/w$ | 0.33 | 3 |
| Ellipsoid tall | $d/w$ | 0.33 | 3 | $h/w$ | 3 | 8 |
| Box short | $d/w$ | 0.33 | 3 | $h/w$ | 0.125 | 0.33 |
| Box normal | $d/w$ | 0.33 | 3 | $h/w$ | 0.33 | 3 |
| Box tall | $d/w$ | 0.33 | 3 | $h/w$ | 3 | 8 |

Table 6: Normalized Mutual Information (NMI), Adjusted Rand Index (ADI), and Silhouette Coefficient (SC) of the vanilla autoencoder and regularized autoencoder

| MODEL | NMI | ADI | SC |
|---|---|---|---|
| Vanilla AE | $0.7624 \pm 0.2132$ | $0.7209 \pm 0.2598$ | $0.4207 \pm 0.0453$ |
| Regularized AE | $\mathbf{0.9484 \pm 0.1391}$ | $\mathbf{0.9368 \pm 0.1737}$ | $\mathbf{0.5279 \pm 0.0817}$ |

higher values, indicating that our regularization technique makes the autoencoder learn more optimal latent spaces.

Some representative examples are shown in Figure 9. The trend of the experimental results is similar to the experimental results in Section 5.1.2. For all five results, the gray ellipses are aligned well with the decision boundary in the vanilla autoencoder (we show the decision boundary in Figure 9 while it is not included in the main manuscript due to lack of space). In the regularized autoencoder, these gray ellipses (or Riemannian metrics) try to become isotropic, so the gaps on the decision boundaries get widened. As a result, different class clusters become farther away from each other. The interpretations about results on GMM and pairwise distances are same as the ones in Section 5.1.2.

### E.1.2 LINEAR INTERPOLATIONS USING REGULARIZED AUTOENCODERS

The generated point clouds from the representative intra-class linear interpolants between two cylinders and two cones with the regularized autoencoders with the Euclidean metric and info-Riemannian metric are drawn in Figure 10.

### E.2 STANDARD BENCHMARK DATASET

### E.2.1 PERFORMANCE ANALYSIS WITH VARYING REGULARIZATION COEFFICIENTS

Figure 11 shows graphs of the classification accuracy versus reconstruction error for the trained AEs measured on ModelNet datasets, for a range of regularization coefficients. The reconstruction error is measured by the modified Chamfer distance as in (Yang et al., 2018b). Compared to vanilla autoencoders (red), regularized autoencoders under the info-Riemannian metric (blue) show overall higher classification accuracy regardless of the regularization coefficients. At the same time, they do not significantly increase the reconstruction error. On the other hand, when comparing the performance of regularized autoencoders under the Euclidean metric (green), most of these are clearly inferior to the vanilla autoencoder; the others perform even worse. Overall, regularization under the info-Riemannian metric is much more robust to the choice of regularization coefficients compared to using the Euclidean metric.

The linear SVM classification accuracy and reconstruction error (modified Chamfer distance) according to regularization coefficient are shown in Table 9 and Table 10. The tables are also arranged

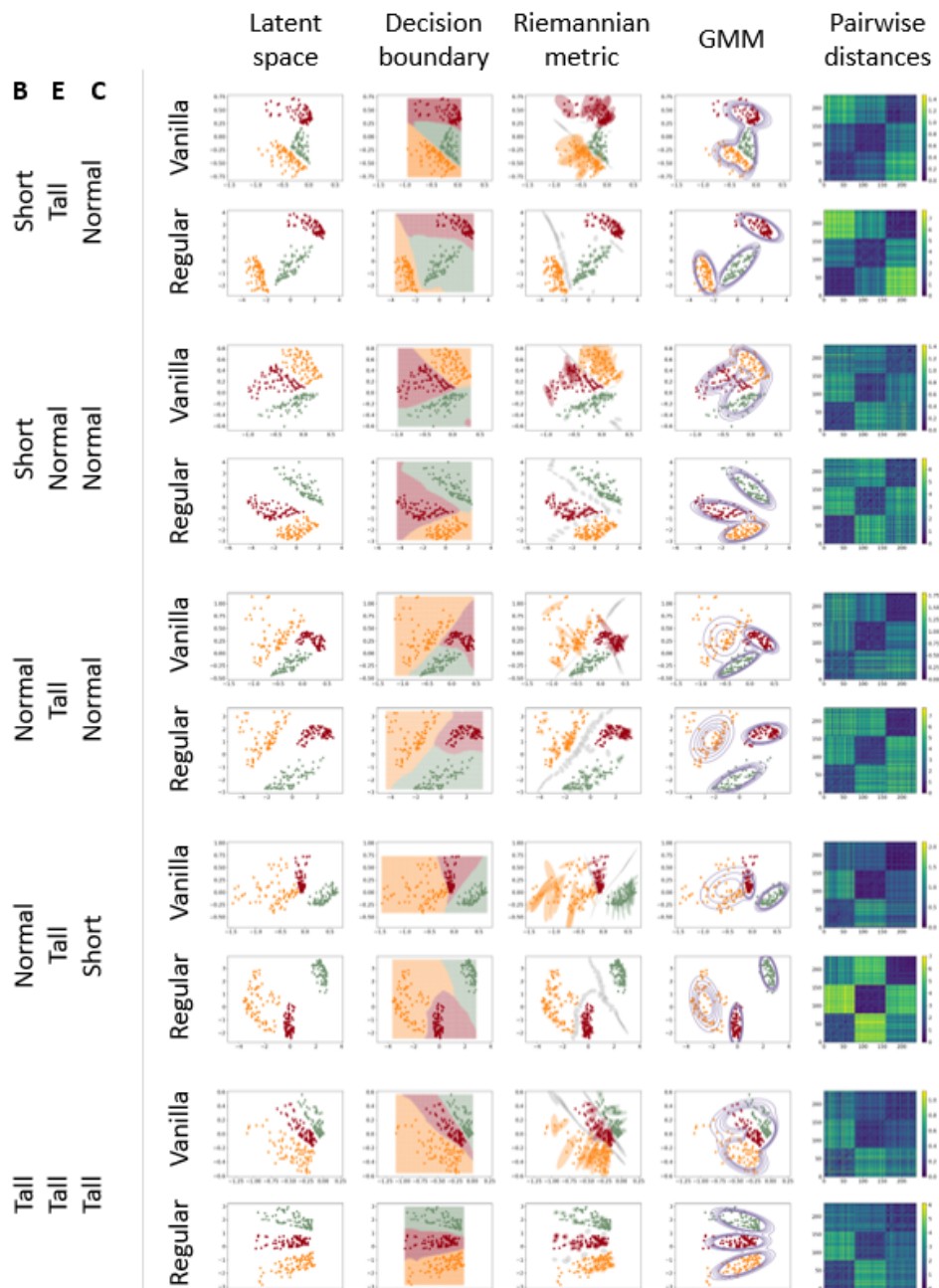

Figure 9: The representative five examples of the regularization experiments on the synthetic dataset. From left to right: latent spaces, decision boundary according to the color assigning method introduced in Appendix D.2, latent spaces with equidistant ellipse ($\{z|(z-z^*)^T G(z^*)(z-z^*) = 1\}$ for center $z^*$) centered on some selected points and sampled points from interspaces, Gaussian Mixture Model (GMM) fitting results, and the heat map of the pairwise Euclidean distances in the latent space of all test data. For each experiment, the upper figure is a vanilla autoencoder trained without regularization, while the lower figure is trained with regularization.

according to autoencoder models (FcNet vs. Foldingnet vs. PointCapsNet vs. DGCNN-FcNet) and regularization types (Vanilla vs. Euclidean vs. info-Riemannian).

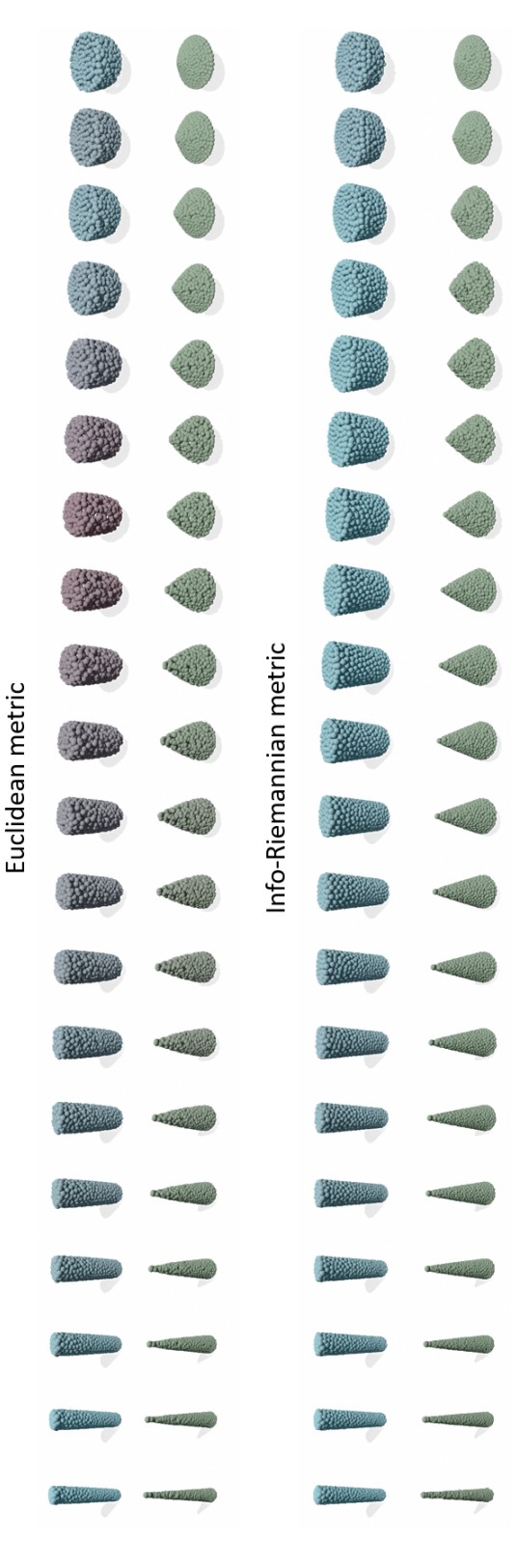

Figure 10: The generated point clouds from the linear interpolants of the regularized autoencoders with the Euclidean metric (*Upper*) and info-Riemannian metric (*Lower*).

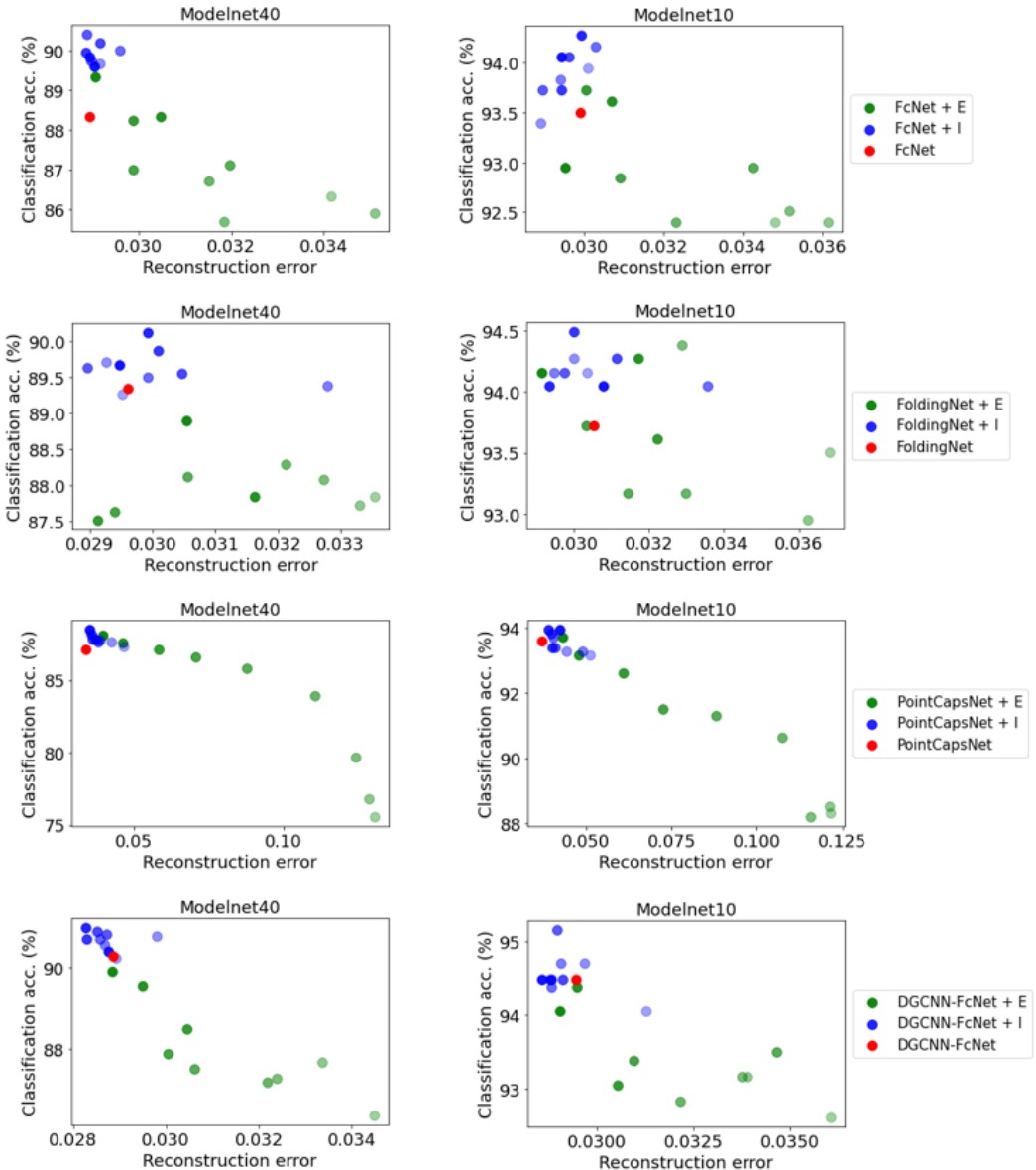

Figure 11: Graphs of classification accuracy versus reconstruction error measured on ModelNet datasets. More transparent markers have larger coefficients $\lambda$; detailed values are in Table 9 and Table 10.

### E.2.2 LEARNING CURVES FOR NOISY POINT CLOUD DATA

Figure 12 shows how the linear SVM classification accuracy and reconstruction error (modified Chamfer distance) evolve as the training proceeds, where datasets are ModelNet10 and ModelNet40 and noise levels are 1%, 5%, 10%, and 20% (details about noise are in Appendix D.4). Compared to vanilla autoencoders (light colored lines), regularized autoencoders under Info-Riemannian metrc (dark colored lines) show overall higher classification accuracy, while they show similar levels of reconstruction errors. The increase in classification accuracy becomes more pronounced as the noise level increases. Especially, when the noise level is 10% and 20%, the classification accuracy of the vanilla autoencoder and regularized autoencoder under Euclidean metric gradually decreases as the learning progresses (as the epoch increases). However, such phenomenons do not appear in the

regularized autoencoders under the Info-Rimennian metric. This result implies that our method is very advantageous in situations where there is noise in the data.

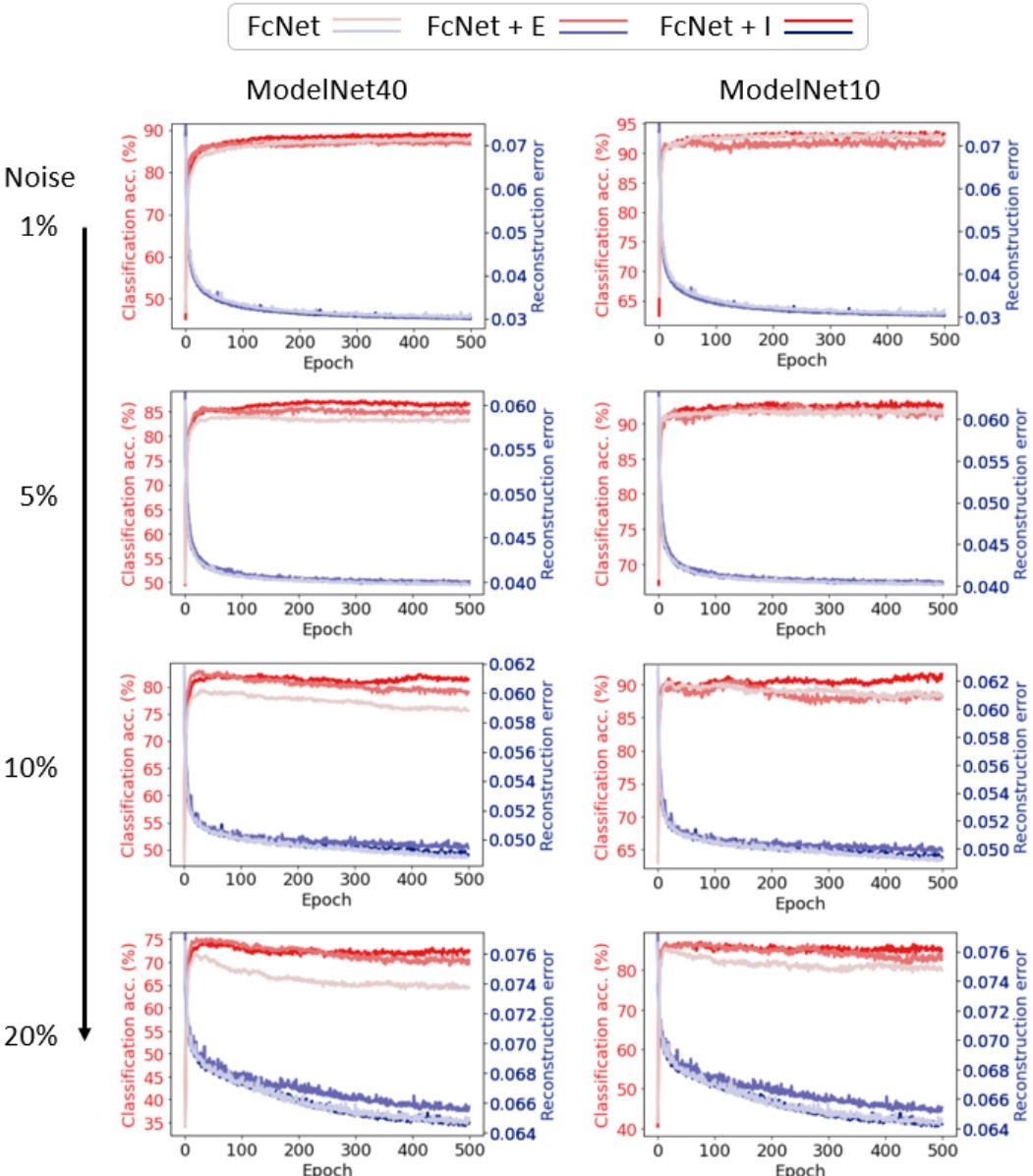

Figure 12: Learning curves of classification accuracy and reconstruction error measured on Model-Net datasets (ModelNet40 and ModelNet10) according to the noise levels (1%, 5%, 10%, and 20%). In each plot, the light colored lines are the result of the non-regularized autoencoders (i.e., FcNet), and the dark colored lines are the result of the regularized autoencoders (i.e., FcNet + E and FcNet + I).

### E.2.3 EXPERIMENTAL RESULTS ON SEMI-SUPERVISED CLASSIFICATION

We train *FcNet* whose the latent space is 512-dimensional with and without regularization. The training configuration is the same with the case of Appendix D.3 except the following: the regularization term of the regularized autoencoder (i.e., FcNet + I) is multiplied by $\lambda = 8000$. In this case, when we training linear SVM classifier, we only use the different numbers of training data (1%, 5%, 10%, and 50% of the overall training data).

Table 7 shows a comparison of transfer classification accuracy according to the percentage of training label used. The overall trend is similar to the results of the experiment with noise in Section 5.2 and Appendix E.2.2. Regularized autoencoders (i.e., FcNet + I) show overall higher classification accuracy compared to vanilla autoencoders (i.e., FcNet), while their reconstruction errors are not significantly different to vanilla autoencoders'. Moreover, the increase in classification accuracy becomes more pronounced as the label rate decreases. In other words, our performance is more effective as the number of labels decreases. Figure 13 shows how the linear SVM classification accuracy and reconstruction error evolve as the training proceeds, where label rate levels are 1%, 5%, 10%, and 50%. Also, similarly, when the label rate is 1%, the classification accuracy of the vanilla autoencoder gradually decreases as the learning progresses (as the epoch increases), but such phenomenons do not appear in the regularized autoencoders. This result implies that our method is also very advantageous in semi-supervised settings.

Table 7: Classification accuracy by transfer learning for ModelNet10 (MN10) and ModelNet40 (MN40) from ShapeNet under the the percentages of labeled training data of linear SVM classifier (50%, 10%, 5%, and 1%).

|                 | MN40 | | | | MN10 | | | |
|-----------------|-------|-------|-------|-------|-------|-------|-------|-------|
|                 | 50%   | 10%   | 5%    | 1%    | 50%   | 10%   | 5%    | 1%    |
| FcNet           | 85.7% | 78.0% | 70.6% | 50.3% | 91.7% | 90.1% | 87.2% | 74.1% |
| FcNet + I (ours)| **87.9%** | **81.6%** | **76.8%** | **57.4%** | **93.2%** | **91.2%** | **88.3%** | **78.1%** |

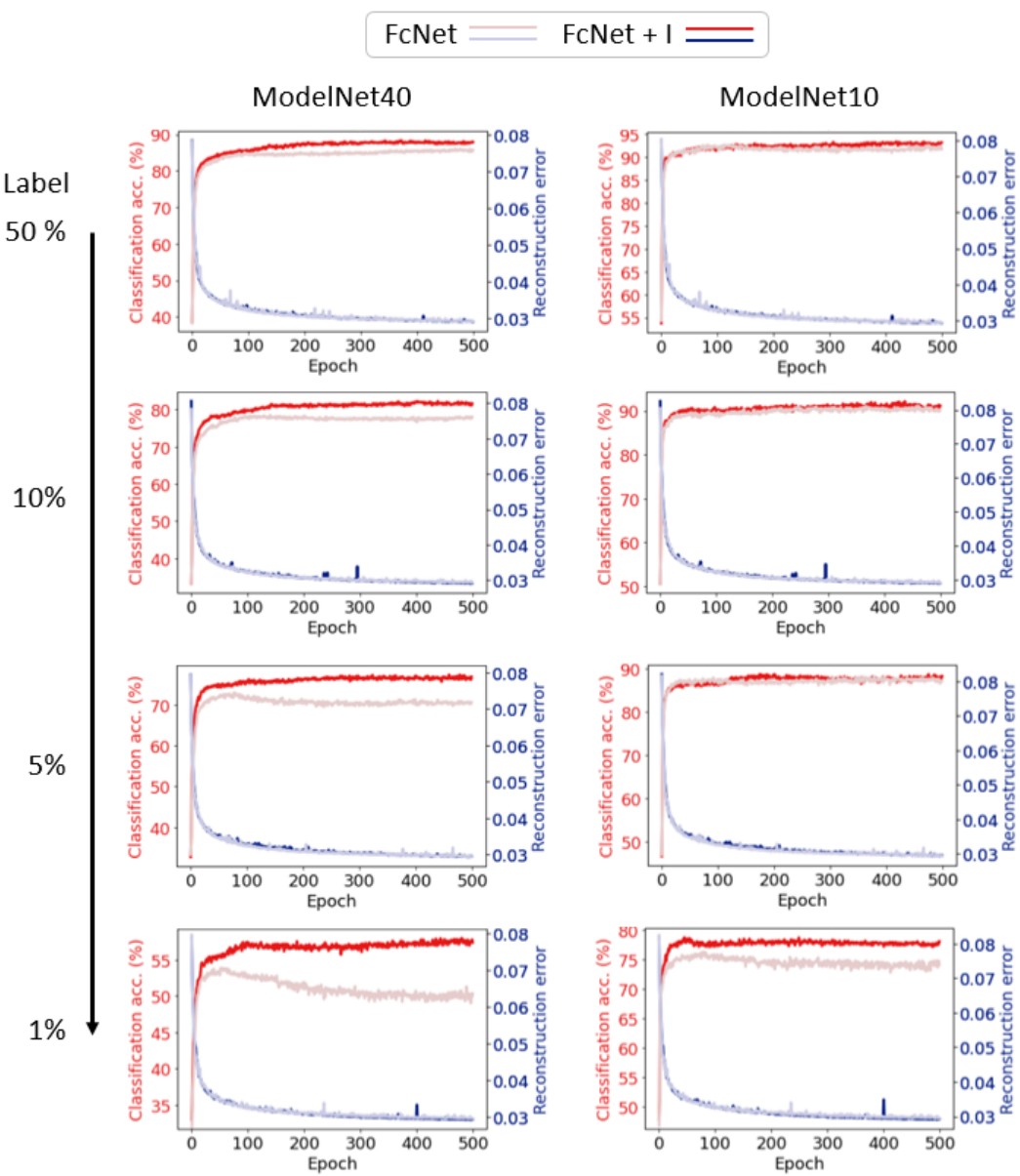

Figure 13: Learning curves of classification accuracy and reconstruction error measured on Model-Net datasets (ModelNet40 and ModelNet10) according to the label rates (50%, 10%, 5%, and 1%). In each plot, the light colored lines are the result of the non-regularized autoencoders (i.e., FcNet), and the dark colored lines are the result of the regularized autoencoders (i.e., FcNet + I).

## F   COMPUTATIONAL ASPECT

First of all, the proposed info-Riemannian metric

$$H_{ijkl}(X) = \sum_r \frac{K\left(\frac{x_r - x_i}{\sigma}\right) K\left(\frac{x_r - x_k}{\sigma}\right)}{\left(\sum_{m=1}^n K\left(\frac{x_r - x_m}{\sigma}\right)\right)^2} \left[\frac{(x_r - x_i)(x_r - x_k)^T}{\sigma^4}\right]_{jl}$$

requires to compute all pairwise distances between points in $X$. This can be computed in parallel using GPU, and it requires a similar level of computation to that of calculating the widely-used average Hausdorff distance metric between point cloud data.

Table 8: Ellapsed time per one epoch when training FcNet, FcNet + E, and FcNet + I

| MODEL | FcNet | FcNet + E | FcNet + I |
|---|---|---|---|
| TIME (s) | 90.97 | 146.27 | 287.51 |

On the other hand, since $H_{ijkl}$ is a huge 4D tensor, there may be a memory issue, but fortunately, there is no need to store this tensor for both autoencoder application tasks studied in this paper: (i) geodesic interpolation and (ii) learning optimal latent space coordinates. Because the calculations commonly required to perform both tasks can be done without storing the tensor $H_{ijkl}$ entirely. Specifically, the commonly-required calculations are as follows: given a neural network decoder $f : \mathbb{R}^m \to \mathbb{R}^{n \times D}$ and a latent value $z \in \mathbb{R}^m$ with tangent vector $v = (v^1, \ldots, v^m) \in \mathbb{R}^m$, we need to compute

$$\sum_{i,j,k,l,a,b} H_{ijkl}(f(z))(J_f)^{ij}_a(z)(J_f)^{kl}_b(z)v^a v^b,$$

where $J_f$ is the Jacobian of $f$. This can be memory-efficiently done as described in the pytorch-style pseudocode:

```python
def Info_Riemannian_velocity_norm(f, z, v, sigma):
    """
    input: decoder function 'f'     # torch.nn.Module, output size:
                                    #   (batch_size, num_pnts, x_dim)
           latent value 'z'         # (batch_size, latent_dim)
           tangent vector 'v'       # (batch_size, latent_dim)
           bandwith 'sigma'         # scalar
    output: velocity square 'v_sq'  # (batch_size)
    """
    # Jacobian vector product
    X, Jv = torch.autograd.functional.jvp(f, z, v)

    # calculate Info-Riemannian metric
    batch_size, num_pts, x_dim = X.size()
    delta_X = (X.unsqueeze(2) - X.unsqueeze(1)) / sigma
    K_delta_X = Kernel(delta_X.view(-1, x_dim))
    K_delta_X = K_delta_X.view(batch_size, num_pts, num_pts)
    K_delta_X = K_delta_X / K_delta_X.sum(dim=2).unsqueeze(2)

    # summation
    v_sq = torch.einsum('nxi, nxij, nij -> nx', K_delta_X, delta_X, Jv)
    v_sq = torch.einsum('nx, nx -> n', v_sq, v_sq)
    return v_sq / (sigma ** 2)

def Kernel(u):
    """
    input: vector 'u'               # (batch_size, vec_dim)
    output: kernel values 'k_u'     # (batch_size)
    """
    vec_dim = u.size(1)
    k_u = 1/(2*pi)**(vec_dim/2) * torch.exp(-torch.norm(u, dim=1)**2/2)
    return k_u
```

Listing 1: Pytorch-style pseudocode for the calculation of velocity square under Info-Riemannian metric

For training the regularized autoencoders, it is not only required to compute the above quantity but also back-propagate through it. In the following, we report how costly adding the regularization term is, compared to the vanilla autoencoder case. We measure the ellapsed time per one training epoch under GeForce RTX 3090. For representative examples, we use FcNet, FcNet + E, and FcNet + I, and the ellapsed tims for each method is shown in Table 8.

## G  DIFFERENT KERNEL CHOICES FOR DENSITY ESTIMATIONS

In this section, we qualitatively analyze how the statistical representation can change when a kernel function other than the Gaussian kernel function is used. As representative ones, we compare the Gaussian, Uniform, Triangular, Epanechnikov, and Sigmoid kernel functions. All but Gaussian and Sigmoid kernel functions have the finite support. For visualization, comparison is made using 2d point clouds, and data in the shape of circles, ellipses, squares, stars, and hearts are used.

Figure 14 visualizes the probability distributions. Basically, the estimated probability distributions are not that different. However, when using kernel functions with finite support, estimated distributions are not very smooth compared to using a Gaussian or Sigmoid kernel function that have infinite support. Overall, it is difficult to say definitively which choice is better, but we choose the one that produces smooth estimations, the Gaussian kernel.

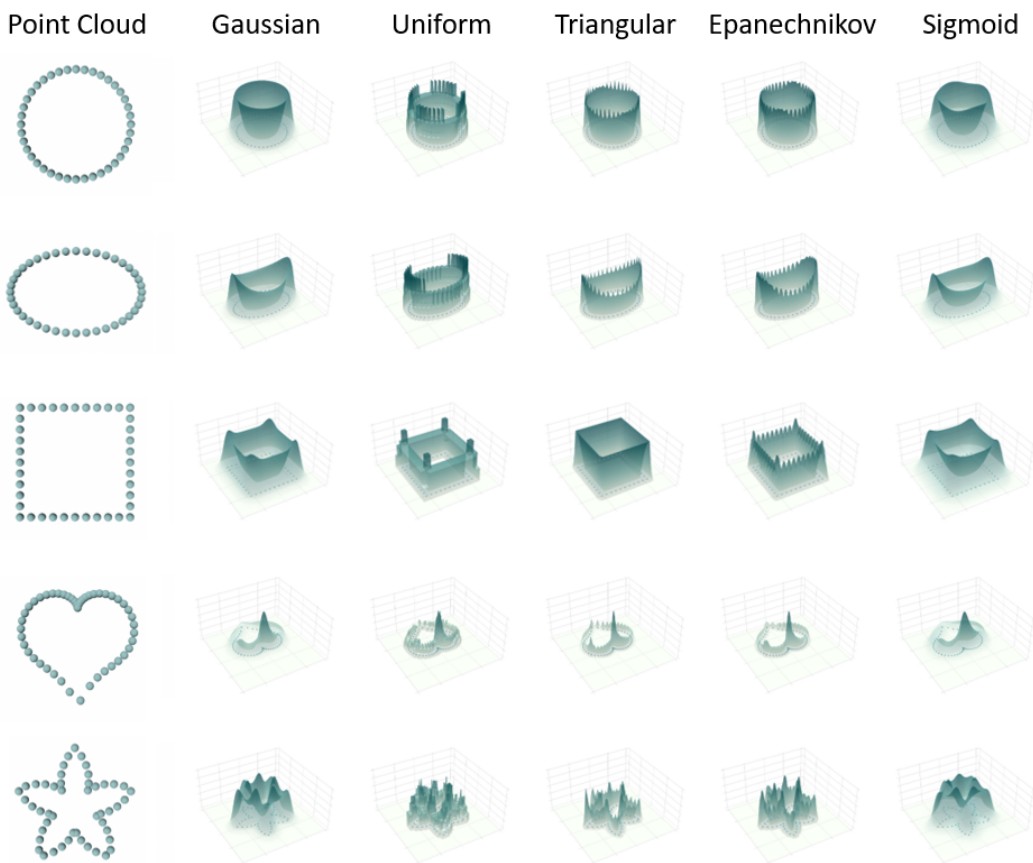

Figure 14: Kernel density functions of different point cloud data (circle, ellipse, square, heart, and star) using different kernels (Gaussian, uniform, triangular, Epanechnikov, and sigmoid). We set the common bandwidth of kernels $\sigma = \text{MED}$, where MED denotes the median of the distances between the points in the point cloud and their nearest points.

Table 9: Classification accuracy and reconstruction error according to regularization coefficient. The table is also arranged according to model (FcNet vs. FoldingNet) and regularization type (Vanilla vs. Euclidean vs. info-Riemannian). For Riemannian metric cases, the regularization coefficients used in the actual experiments are $\sigma^2$ times $\lambda$ shown in the table.

| MODEL | METRIC | $\lambda$ | MD40 acc | MD40 recon | MD10 acc | MD10 recon |
|---|---|---|---|---|---|---|
| FcNet | Euclidean | 0.0001 | 89.343598 | 0.029069 | 92.951542 | 0.029527 |
| FcNet | Euclidean | 0.0010 | 88.330632 | 0.030464 | 93.612335 | 0.030684 |
| FcNet | Euclidean | 0.0100 | 88.249595 | 0.029877 | 93.722467 | 0.030054 |
| FcNet | Euclidean | 0.1000 | 86.993517 | 0.029878 | 92.841410 | 0.030900 |
| FcNet | Euclidean | 1.0000 | 87.115073 | 0.031966 | 92.951542 | 0.034272 |
| FcNet | Euclidean | 10.0000 | 86.709887 | 0.031523 | 92.400881 | 0.032318 |
| FcNet | Euclidean | 100.0000 | 85.696921 | 0.031841 | 92.511013 | 0.035172 |
| FcNet | Euclidean | 1000.0000 | 85.899514 | 0.035089 | 92.400881 | 0.036145 |
| FcNet | Euclidean | 10000.0000 | 86.345219 | 0.034149 | 92.400881 | 0.034811 |
| FcNet | Riemannian | 100.0000 | 89.586710 | 0.029032 | 94.052863 | 0.029435 |
| FcNet | Riemannian | 500.0000 | 89.829822 | 0.028944 | 94.273128 | 0.029928 |
| FcNet | Riemannian | 1000.0000 | 89.951378 | 0.028864 | 93.722467 | 0.029430 |
| FcNet | Riemannian | 2000.0000 | 90.194489 | 0.029165 | 94.052863 | 0.029606 |
| FcNet | Riemannian | 8000.0000 | 90.397083 | 0.028869 | 93.722467 | 0.028944 |
| FcNet | Riemannian | 10000.0000 | 89.991896 | 0.029603 | 94.162996 | 0.030278 |
| FcNet | Riemannian | 20000.0000 | 89.748784 | 0.028980 | 93.832599 | 0.029412 |
| FcNet | Riemannian | 50000.0000 | 89.667747 | 0.029161 | 93.392070 | 0.028894 |
| FcNet | Riemannian | 100000.0000 | 89.748784 | 0.028961 | 93.942731 | 0.030085 |
| FcNet | Vanilla | 0.0000 | 88.330632 | 0.028938 | 93.502203 | 0.029895 |
| FoldingNet | Euclidean | 0.0001 | 88.897893 | 0.030540 | 94.162996 | 0.029130 |
| FoldingNet | Euclidean | 0.0010 | 87.844408 | 0.031623 | 94.273128 | 0.031726 |
| FoldingNet | Euclidean | 0.0100 | 87.520259 | 0.029126 | 93.612335 | 0.032219 |
| FoldingNet | Euclidean | 0.1000 | 87.641815 | 0.029391 | 93.722467 | 0.030318 |
| FoldingNet | Euclidean | 1.0000 | 88.128039 | 0.030564 | 93.171806 | 0.031435 |
| FoldingNet | Euclidean | 10.0000 | 88.290113 | 0.032129 | 93.171806 | 0.032975 |
| FoldingNet | Euclidean | 100.0000 | 88.087520 | 0.032728 | 94.383260 | 0.032885 |
| FoldingNet | Euclidean | 1000.0000 | 87.722853 | 0.033305 | 92.951542 | 0.036244 |
| FoldingNet | Euclidean | 10000.0000 | 87.844408 | 0.033546 | 93.502203 | 0.036820 |
| FoldingNet | Riemannian | 100.0000 | 89.667747 | 0.029467 | 94.052863 | 0.030789 |
| FoldingNet | Riemannian | 500.0000 | 90.113452 | 0.029916 | 94.052863 | 0.029346 |
| FoldingNet | Riemannian | 1000.0000 | 89.870340 | 0.030090 | 94.493392 | 0.029996 |
| FoldingNet | Riemannian | 2000.0000 | 89.546191 | 0.030471 | 94.273128 | 0.031142 |
| FoldingNet | Riemannian | 8000.0000 | 89.627229 | 0.028949 | 94.162996 | 0.029745 |
| FoldingNet | Riemannian | 10000.0000 | 89.505673 | 0.029926 | 94.052863 | 0.033548 |
| FoldingNet | Riemannian | 20000.0000 | 89.384117 | 0.032798 | 94.162996 | 0.029463 |
| FoldingNet | Riemannian | 50000.0000 | 89.708266 | 0.029262 | 94.273128 | 0.030002 |
| FoldingNet | Riemannian | 100000.0000 | 89.262561 | 0.029511 | 94.162996 | 0.030355 |
| FoldingNet | Vanilla | 0.0000 | 89.343598 | 0.029599 | 93.722467 | 0.030528 |

Table 10: Classification accuracy and reconstruction error according to regularization coefficient. The table is also arranged according to model (PointCapsNet vs. DGCNN-FcNet) and regularization type (Vanilla vs. Euclidean vs. info-Riemannian). For Riemannian metric cases, the regularization coefficients used in the actual experiments are $\sigma^2$ times $\lambda$ shown in the table.

| MODEL | METRIC | $\lambda$ | MD40 acc | MD40 recon | MD10 acc | MD10 recon |
|---|---|---|---|---|---|---|
| PointCapsNet | Euclidean | 0.0001 | 88.087520 | 0.039522 | 93.722467 | 0.043112 |
| PointCapsNet | Euclidean | 0.0010 | 87.601297 | 0.046227 | 93.171806 | 0.047749 |
| PointCapsNet | Euclidean | 0.0100 | 87.155592 | 0.058326 | 92.621145 | 0.060827 |
| PointCapsNet | Euclidean | 0.1000 | 86.628849 | 0.070735 | 91.519824 | 0.072615 |
| PointCapsNet | Euclidean | 1.0000 | 85.858995 | 0.087722 | 91.299559 | 0.087930 |
| PointCapsNet | Euclidean | 10.0000 | 83.954619 | 0.110708 | 90.638767 | 0.107402 |
| PointCapsNet | Euclidean | 100.0000 | 79.659643 | 0.124266 | 88.215859 | 0.115549 |
| PointCapsNet | Euclidean | 1000.0000 | 76.823339 | 0.128694 | 88.546256 | 0.121284 |
| PointCapsNet | Euclidean | 10000.0000 | 75.567261 | 0.130566 | 88.325991 | 0.121398 |
| PointCapsNet | Riemannian | 100.0000 | 88.492707 | 0.035034 | 93.942731 | 0.042224 |
| PointCapsNet | Riemannian | 1000.0000 | 88.168558 | 0.035816 | 93.942731 | 0.039131 |
| PointCapsNet | Riemannian | 2000.0000 | 87.884927 | 0.036824 | 93.392070 | 0.040169 |
| PointCapsNet | Riemannian | 5000.0000 | 87.884927 | 0.035990 | 93.832599 | 0.039729 |
| PointCapsNet | Riemannian | 8000.0000 | 87.641815 | 0.037986 | 93.392070 | 0.040947 |
| PointCapsNet | Riemannian | 10000.0000 | 87.763371 | 0.037507 | 93.722467 | 0.040428 |
| PointCapsNet | Riemannian | 20000.0000 | 87.763371 | 0.038454 | 93.281938 | 0.048909 |
| PointCapsNet | Riemannian | 50000.0000 | 87.641815 | 0.042233 | 93.281938 | 0.044333 |
| PointCapsNet | Riemannian | 100000.0000 | 87.317666 | 0.046607 | 93.171806 | 0.051187 |
| PointCapsNet | Vanilla | 0.0000 | 87.155592 | 0.033936 | 93.612335 | 0.037172 |
| DGCNN-FcNet | Euclidean | 0.0001 | 89.910859 | 0.028852 | 94.052863 | 0.029049 |
| DGCNN-FcNet | Euclidean | 0.0010 | 89.546191 | 0.029489 | 94.383260 | 0.029480 |
| DGCNN-FcNet | Euclidean | 0.0100 | 88.492707 | 0.030448 | 93.061674 | 0.030531 |
| DGCNN-FcNet | Euclidean | 0.1000 | 87.884927 | 0.030046 | 93.392070 | 0.030966 |
| DGCNN-FcNet | Euclidean | 1.0000 | 87.520259 | 0.030605 | 93.502203 | 0.034671 |
| DGCNN-FcNet | Euclidean | 10.0000 | 87.196110 | 0.032174 | 92.841410 | 0.032151 |
| DGCNN-FcNet | Euclidean | 100.0000 | 87.277147 | 0.032385 | 93.171806 | 0.033750 |
| DGCNN-FcNet | Euclidean | 1000.0000 | 87.682334 | 0.033359 | 93.171806 | 0.033906 |
| DGCNN-FcNet | Euclidean | 10000.0000 | 86.385737 | 0.034471 | 92.621145 | 0.036056 |
| DGCNN-FcNet | Riemannian | 100.0000 | 90.397083 | 0.028761 | 94.493392 | 0.028591 |
| DGCNN-FcNet | Riemannian | 1000.0000 | 90.964344 | 0.028278 | 94.493392 | 0.028806 |
| DGCNN-FcNet | Riemannian | 2000.0000 | 90.680713 | 0.028299 | 94.493392 | 0.028832 |
| DGCNN-FcNet | Riemannian | 5000.0000 | 90.883306 | 0.028520 | 94.493392 | 0.029128 |
| DGCNN-FcNet | Riemannian | 8000.0000 | 90.802269 | 0.028727 | 95.154185 | 0.028971 |
| DGCNN-FcNet | Riemannian | 10000.0000 | 90.680713 | 0.028583 | 94.383260 | 0.028820 |
| DGCNN-FcNet | Riemannian | 20000.0000 | 90.559157 | 0.028677 | 94.713656 | 0.029067 |
| DGCNN-FcNet | Riemannian | 50000.0000 | 90.761750 | 0.029794 | 94.713656 | 0.029695 |
| DGCNN-FcNet | Riemannian | 100000.0000 | 90.235008 | 0.028934 | 94.052863 | 0.031275 |
| DGCNN-FcNet | Vanilla | 0.0000 | 90.275527 | 0.028867 | 94.493392 | 0.029454 |

