# OpenReview forum: "A Statistical Manifold Framework for Point Cloud Data"
_ICLR.cc/2022/Conference — ICLR 2022 Submitted_

### Official Review · Reviewer_yY6L · 2021-10-27

**Correctness:** 4
**Technical Novelty And Significance:** 3
**Empirical Novelty And Significance:** 2
**Recommendation:** 5
**Confidence:** 4

**Main Review:**

===STRENGTHS===

NOVELTY

The proposed approach seems novel, and I am not aware of other works that propose a similar approach to equip a point cloud of a Fisher Information Metric.

CLARITY

The paper is clear, and I had no trouble understanding it. The only parts a bit obscure to me are the paragraphs of Section 4; I would suggest giving the applications a bit more space. For example: how is implemented and solved the optimization problem of "Geodesic Interpolation"?

SOUNDNESS AND REPLICABILITY

By my understanding, the method seems sound; while I have not performed an in-depth check, the passages seem reasonable to me. The appendix provides information to help the replicability of the results, explaining the implementation details.


===WEAKNESSES===

EXPERIMENTS

My major concern about this paper is the experimental setting for the following reasons:\
a) Quantitative results are provided only on one experiment of shape classification; in this experiment, the comparisons with other methods are "adopted from references", and hence they are not directly comparable. I am convinced that the proposed method work in a similar level of performance, but at the moment, some sentences seem over-claims not well-grounded (e.g., "the classification accuracy of our method is higher in most cases"). The improvement over the vanilla baselines seems tight; can you provide some elaboration on the "best" cases, i.e., in which cases using the regularization produce the greatest improvement?\
Concretely: Given the tight margin between the proposed method and the ones taken from other papers, I think that it would be important to reproduce at least two SOTA methods (e.g., PointCaps and Multi-Task) in the exact same setting.\
b) The toy dataset is really interesting from the point of analysis, but only qualitative examples are provided. I would also suggest providing some quantitative results in this case that are highly controllable and naturally in correspondence, providing visualization of the reconstruction error. This would give insight into the proposed metric's behavior on different geometrical features (e.g., different curvature and sharpness).\
c) Only rigid objects have been shown, all with the same distributions, while one of the paper's claims is that defining probability distribution is "likely quite natural and intuitive to the user". I would suggest proposing at least some qualitative examples on different domains and different probability distributions. Are there issues in considering non-rigid objects? Which are other reasonable choices for probability distributions that respect the given assumptions?

CONTEXT

The paper does not offer a proper context around the work. I would suggest providing at least two sections for related works: one on point clouds as a probability distribution (as stated in the text, this is not a novel idea), and one for defining metrics for point clouds and their latent representations (examples are [1] and [2]). I think that the correct positioning of the paper in the literature would help to clarify the novelty of the work and what it is actually proposing.


===MINOR ISSUES AND QUESTIONS===

1) the "info-Riemannian" name is introduced twice (page 2 and page 4)
2) to me it is not completely obvious the term "velocity" on page 4; could you give me a better intuition\definition of it?
3) I would suggest coloring the point clouds of Figure 4 with a non-constant color to better track the movement of the points; I see the overall structure is better "preserved" in IR metric, but I would better understand their movements.
4) the computational aspect seems untouched; can you provide some timings of your computation, e.g., comparing it with the geodesic metric computation over a mesh?

[1]: Limp: Learning latent shape representations with metric preservation priors, Cosmo et al., 2020

[2]: Point-set Distances for Learning Representations of 3D Point Clouds, Nguyen et al., 2021

**Summary Of The Paper:**

The paper proposes treating the point clouds as probability distributions and equipping them with a Fisher Information Metric, providing a Riemannian geometric structure that can be used as regularization in AutoEncoders for training and latent interpolation. The experiments are performed on ModelNet and a toy dataset of toy shapes (cones, cylinders, ellipsoids).

**Summary Of The Review:**

I like the overall idea, and some qualitative results seem promising. However, some aspects require better experimentation and remain a bit obscure, particularly the effectiveness of the method, the positioning among other SOTA approaches, and the behavior in different conditions (e.g., different domains, probability distribution).

---

> ### Author Response · Authors · 2021-11-18
> **Response to Reviewer yY6L (4/4)**
>
> __Q7.__ To me it is not completely obvious the term "velocity"
> on page 4; could you give me a better intuition/definition of it?
>
> __A7.__ Sure. Consider a point cloud $\mathbf{X}:=${$x_1,\dots,x_n|x_i\in
> \mathbb{R}^2$} and velocity set $\mathbf{V}:=${$v_1,\dots,v_n|v_i\in
> \mathbb{R}^2$} such that $v_i$ are velocity vectors of the points $x_i$.
> One obvious (but naive) way of defining the norm of $\mathbf{V}$ is
> $$\sum_{i=1}^{n} \|v_i\|^2,$$
> which is identical to the expression in page 4, $\sum_i \sum_j V^{ij} V^{ij}$.
> This Euclidean norm considers each point as uncorrelated and does not
> capture the distributional property of the point cloud. As explained
> in the main manuscript (Figure 3), our info-Riemannian metric produces
> a more reasonable norm for $\mathbf{V}$.  We hope this is of some help.
>
> __Q8.__ The computational aspect seems untouched; can you provide
> some timings of your computation, e.g., comparing it with the geodesic
> metric computation over a mesh?
>
> __A8.__ Thank you for the suggestion.
> ``We have added a discussion on the computational aspects of our
> method in Appendix F``.  First, we provide details on how the metric can be
> computed in a memory-efficient way, with pytorch-style pseudocode added.
> Second, we have provided the per-epoch running times of autoencoder
> training both with and without regularization.
>
> __Q9.__ I would suggest coloring the point clouds of Figure 4 with
> a non-constant color to better track the movement of the points; I see
> the overall structure is better "preserved" in IR metric, but I would
> better understand their movements.
>
> __A9.__ Thank you for this suggestion. ``We now color
> the point clouds of Figure 4 with a non-constant color.`` As the reviewer
> suggested, after changing to non-constant color, we were able to better
> track the movement of the points, so that it is visually easier to see that
> the random walk under the Info-Riemannian metric preserves well the overall
> structure of the point cloud.
>
> __Q10.__The "info-Riemannian" name is introduced twice (page 2 and page 4)
>
> __A10.__ Thank you for pointing out this redundancy.
> ``We have removed the redundant introduction of
> ``info-Riemannian" on page 4.``

---

> ### Author Response · Authors · 2021-11-18
> **Response to Reviewer yY6L (3/4)**
>
> __Q4.__ The toy dataset is really interesting from the point of analysis, but only qualitative examples are provided. I would also suggest providing some quantitative results in this case that are highly controllable and naturally in correspondence, providing visualization of the reconstruction error. This would give insight into the proposed metric's behavior on different geometrical features (e.g., different curvature and sharpness).
>
> __A4.__ Thank you for the suggestion. We agree that providing some
> quantitative results on synthetic experiments is of value. {\color{blue}
> In this regard, we have added Appendix E.1.1 (Quantitative Results on
> Synthetic Data).}
>
> In short, we add some quantitative results focused especially on how much
> clustering accuracy is improved with autoencoder regularization. For
> this additional experiment, we generate sufficiently diverse synthetic
> datasets, and train autoencoders with and without regularization.
> We perform latent space clustering for autoencoders trained on each
> dataset, and measure the clustering score (i.e., normalized mutual
> information).  The experiments provide further quantitative evidence
> that the performance of our method exceeds that of vanilla autoencoders.
> Further details are given in Appendix E.1.1 (Quantitative Results on
> Synthetic Data). We hope that this additional section better
> illuminates how our proposed metric behaves for autoencoder
> regularization tasks.
>
> __Q5.__ The only parts a bit obscure to me are the paragraphs
> of Section 4; I would suggest giving the applications a bit more space.
> For example: how is implemented and solved the optimization problem of
> ``Geodesic Interpolation"?
>
> __A5.__ ``We have added more implementation details
> in Appendix D.2 and Appendix F.`` In Appendix D.2, we add implementation
> details about the geodesic interpolation task. We are aware that the
> current draft is somewhat lacking in information to reproduce geodesic
> interpolation tasks; we now add full details of geodesic interpolation,
> including how the optimization is solved. In Appendix F, we add
> implementation details on how to calculate the proposed info-Riemannian
> metric. We provide pytorch-style pseudocode for a memory-efficient
> implementation for calculating the pullback metric in the latent space.
>
> __Q6.__ Only rigid objects have been shown, all with the same
> distributions, while one of the paper's claims is that defining probability
> distribution is "likely quite natural and intuitive to the user".
> I would suggest proposing at least some qualitative examples on
> different domains and different probability distributions. Are
> there issues in considering non-rigid objects? Which are other
> reasonable choices for probability distributions that respect
> the given assumptions?
>
> __A6.__ Thank you for pointing out these issues. First, in the revised paper,
> ``we have added in Appendix G a qualitative analysis of how
> the statistical representation changes when a kernel function other than
> the Gaussian kernel function is used.``
>
> In Appendix G, we compare the Gaussian, Uniform, Triangular, Epanechnikov,
> and Sigmoid kernel functions; for visualization purposes, comparisons are
> made with 2d point clouds: circles, ellipses, squares, stars, and hearts.
> In short, kernel functions with infinite support (e.g., Gaussian and sigmoid) produce
> smooth estimates, whereas those with finite support (e.g., uniform,
> triangular, and Epanechnikov) produce non-smooth estimates.
> The overall shape is similar, but each has its own characteristics, so
> the optimal kernel choice should depend on the task and dataset. Although
> we mainly focus on the Gaussian kernel in the paper, testing our algorithm
> with other kernel functions and comparing the results could be
> an interesting future research direction.
>
> Second, we believe there is no major difficulty in applying our framework
> to non-rigid objects.  In fact, our Riemannian manifold framework is just
> the right tool to analyze how non-rigid objects evolve or deform over time,
> since by using our proposed Riemannian metric, we can measure dynamic
> aspects like its velocity, which was not possible within existing frameworks.
> This is a very insightful observation made by reviewer. We definitely
> plan to apply our method in applications involving non-rigid objects (in
> fact, we have such an application in mind for robot grasping).

---

> ### Author Response · Authors · 2021-11-18
> **Response to Reviewer yY6L (2/4)**
>
> __Q2.__ Given the tight margin, I think that it would be
> important to reproduce at least two SOTA methods (e.g., PointCaps
> and Multi-Task).
>
> __A2.__ We agree with the reviewer’s point. ``We
> now have reproduced recent SOTA methods, PointCapsNet and
> DGCNN-FCNet, and added the classification accuracy results in Table 1.``
>
> From the algorithms existing in ``adopted from references" in the paper, we
> select PointCapsNet, a paper that performs pure autoencoding (i.e.,
> training using only the reconstruction loss as the training loss), as
> one SOTA method. Also, inspired from experiments using synthetic data
> (Section 5.1), we empirically find that DGCNN-FcNet shows good
> classification accuracy. DGCNN-FcNet is therefore adopted as another SOTA method.
>
> Consistent with previous results, adding our regularization term improves
> classification accuracy in a meaningful way. Further information about
> the newly SOTA methods and analysis of experimental results is contained
> in Appendix D.3 and E.2.1 (Performance Analysis with Varying Regularization
> Coefficients), respectively.
>
> __Q3.__ I am convinced that the proposed method work in a similar level of performance, but at the moment, some sentences seem over-claims not well-grounded. The improvement over the vanilla baselines seems tight; can you provide some elaboration on the "best" cases, i.e., in which cases using the regularization produce the greatest improvement?
>
> __A3.__ Thank you for raising this issue. First, we agree that some
> sentences describing the current experimental results could have been better phrased.
> In the revised paper, ``we have modified the expression pointed out``.
> Second and  more importantly, we appreciate the reviewer’s question on "when
> using the regularization produces the greatest improvement.''  ``We
> find that using our regularization technique produces the greatest improvements
> (by a significant margin, we might add) in situations where there is noise in
> the point cloud dataset (Appendix E.2.2), and in semi-supervised classification
> settings (Appendix E.2.3).``
>
> We have now added experimental results for noisy point cloud
> data in Section 5.2 and Appendix E.2.2. We conduct additional experiments
> to determine the extent to which robustness can be increased using
> the representation obtained with our regularization method.
> We add different levels of noise (1\%, 5\%, 10\%, and 20\%) to each point
> cloud data, and compare classification accuracy for the vanilla autoencoder
> and the regularized
> autoencoder. More information about the experiments can be found in
> Appendix D.4 (Details for Experiments on Standard Benchmark Dataset with Noise).
> For convenience, we have attached below the performance table now included in the
> revised manuscript.
>
> |                      |        |  MN  |  40     |       |       |   MN    |  10     |       |
> |------------------|-------|--------|--------|-------|-------|-------|-------|-------|
> | Method           | 1%    | 5%    | 10%   | 20%   | 1%    | 5%    | 10%   | 20%   |
> | FcNet            | 87.8% | 83.2% | 75.6% | 64.5% | 92.4% | 91.9% | 88.4% | 79.8% |
> | FcNet + E (ours) | 86.6% | 85.1% | 79.1% | 70.4% | 92.2% | 91.1% | 88.2% | 82.6% |
> | __FcNet + I (ours)__ | __89.0%__ | __86.6%__ | __81.4%__ | __72.4%__ | __93.3%__ | __92.6%__ | __91.6%__ | __84.8%__ |
>
> Compared to vanilla autoencoders (FcNet), regularized autoencoders (FcNet + I)
> show overall higher classification accuracy, with the increase in classification
> accuracy becoming more pronounced with increasing noise levels.
> Another interesting finding from our experimental results concerns
> the learning curve. When the noise level is 10\% and 20\%, the classification
> accuracy of the vanilla autoencoder gradually decreases as learning progresses
> (i.e., with increasing epoch), but this is not the case for regularized autoencoders.
> We refer to Appendix E.2.2 (Learning Curves for Noisy Point Cloud Data) for
> additional information.  In summary, our method shows significant improvements
> over the vanilla baselines in situations where noise is added to the point cloud data.
>
> Moreover, our regularized autoencoders (i.e., FcNet + I) also show higher classification accuracy in semi-supervised setting, and the gap in classification accuracy becomes even more pronounced with decreasing number of labeled training data.
> Please refer to Appendix E.2.3 (Experimental Results on Semi-supervised
> Classification) for experimental settings and more results including the learning curves.
>
> |                      |        |  MN  |  40     |       |       |   MN    |  10     |       |
> |------------------|-------|--------|--------|-------|-------|-------|-------|-------|
> | Method                       | 50%    | 10%    | 5%   | 1%   | 50%    | 10%    | 5%   | 1%   |
> | FcNet                         | 85.7% | 78.0% | 70.6% | 50.3% | 91.7% | 90.1% | 87.2% | 74.1% |
> | __FcNet + I (ours)__ | __87.9%__ | __81.6%__ | __76.8%__ | __57.4%__ | __93.2%__ | __91.2%__ | __88.3%__ | __78.1%__ |

---

> ### Author Response · Authors · 2021-11-18
> **Response to Reviewer yY6L (1/4)**
>
> Thank you very much for your constructive feedback, and our sincerest
> apologies for not uploading our comments sooner. In response to the
> many constructive suggestions we have received, we have spent the
> past week conducting a wide range of new experiments, and revising
> our manuscript accordingly. In an attempt to answer the reviewer
> questions, and to better clarify and validate our contributions,
> the appendix has been substantially expanded with the following
> additional content:
>
> * we have revised the introduction so that it introduces some related/existing works and explains what our contributions are in those contexts;
> * we have added additional results with SOTA autoencoder models in Table 1 in Section 5.2;
> * we have added experiments on ``noisy" point cloud data in Section 5.2, which demonstrates the advantages of our method more clearly;
> * we have added Appendix A (Existing Geometric/Statistical Methods for Point Cloud Data);
> * we have added some more details of algorithms in Appendix D;
> * we have added Appendix E.1.1 (Quantitative Results on Synthetic Data);
> * we have added Appendix E.2.1 (Performance Analysis with Varying Regularization Coefficients);
> * we have added Appendix E.2.2 (Learning Curves of Noisy Point Cloud Data);
> * we have added Appendix E.2.3 (Experimental Results on Semi-supervised Classification);
> * we have added Appendix F (Computational Aspect);
> * we have added Appendix G (Different Kernel Choices for Density Estimations).
>
> Below we provide detailed responses to each of the reviewer comments.
> When referencing any major changes and addition of new content to the
> revised manuscript, we have indicated those passages ``like this``.
>
> __Q1.__ The paper does not offer a proper context around the work. I would suggest providing at least two sections for related works: one on point clouds as a probability distribution (as stated in the text, this is not a novel idea), and one for defining metrics for point clouds and their latent representations (examples are [1] and [2]). I think that the correct positioning of the paper in the literature would help to clarify the novelty of the work and what it is actually proposing.
>
> __A1.__
> We agree that references on previous related studies were lacking, and
> fully agree with the reviewer's opinion that a better positioning of our
> paper in the context of existing works will help clarify the novelty of
> our work. For works on "point clouds as a probability distribution",
> ``we have revised the introduction so that it includes
> existing related works and explains what our contributions are in
> those contexts. Additionally, more detailed discussion on why our
> proposed method is novel vis-a-vis existing geometric/statistical
> methods has been added in Appendix A.``
>
> We agree that the statistical interpretation of point cloud data has
> been explored in some depth in the literature.  Nevertheless, we
> do believe there is clear and meaningful novelty in our approach,
> which we have tried to clarify in the revised version.  As a brief
> summary of our contributions, (i) we rigorously construct
> the statistical manifold structure of point cloud data using a 1-1
> mapping, and (ii) based on the constructed 1-1 mapping and the statistical
> manifold framework, we define a Riemannian manifold structure, and
> obtain an analytic form of the Riemannian metric that can actually
> be used in practice.
>
> In Appendix A, we provide a more in-depth review of existing
> geometric/statistical representations for point cloud data. In short,
> this section includes a "geometric method'' section (A.1) and a
> "statistical method'' section (A.2).  In the former
> we review some popular existing distance metrics for point cloud data
> including the suggested references [1] and [2],
> then explain the limitations of having only distance metric
> to work with, and finally highlight the importance of our Riemannian
> manifold framework.  In the statistical method section, we review
> statistical representations previously used
> for point cloud registration and denoising, and explain illuminate
> the differences between these studies and ours.

---

### Official Review · Reviewer_PyPd · 2021-11-01

**Correctness:** 3
**Technical Novelty And Significance:** 3
**Empirical Novelty And Significance:** 2
**Recommendation:** 6
**Confidence:** 3

**Main Review:**

Strengths:
- the proposed framework has potential, it may indeed sometimes (although certainly not in general) be useful to think about point clouds in this way
- the framework definitely has a certain mathematical elegance
- experiments in toy data like parametric primitives, Shapenet and Modelnet indicate that for "perfect" point clouds the proposed kernel density model is a valid regulariser
- actually the most promising bit of the proposed view for me: an interesting way to construct a regulariser for (global) point cloud representations, although the paper only scratches the surface and does not deeply explore that aspect

Weaknesses:
- the work is in my view interesting, but not as "rigorous" and momentous as claimed; actually, behind the information-theoretic smoke screen it largely boils down to the popular and obvious standard assumption that a point cloud is a set of samples from an underlying surface, perturbed by (with the proposed kernel, Gaussian) random noise
- the view as a distribution may be useful in some situations - typically when requiring a "rough, global" descriptor of the point cloud, for instance to recognise well-discriminated shape categories; it is almost certainly not a suitable view for many other (arguably more realistic and relevant) tasks - for instance the kernel smoothing will harm any local feature extraction or description; and it will likely also be challenged by point clouds that cover the object of interest incompletely or with varying density - in practice any point cloud that has been sensed, rather than rendered synthetically from a surface model (i.e., from an "explicit version of the distribution")
- nothing is said to substantiate the claim that choosing the distribution is easier than making other reasonable assumptions; I am not sure how to make a reasonable choice for realistic distributions (say, autonomous vehicle Lidar)
- the kernel density flavour suggests that the method might not scale all that well to realistic point cloud sizes. Perhaps it does, but the experimental setting with <2100 points per object evades the topic

Comment (neutral, I would not want to call this a strength or a weakness):
- the paper sits in a strange place between a theory paper and a practical, algorithmic machine learning paper: a theory paper can be very valuable, but for my taste the theoretical "contribution" of the paper is fairly basic, it would have been nice to push further and see what more advanced theoretical analyses and conclusions the point cloud manifold view leads to; on the other hand, the practical "evidence" is limited to last-decade toy point clouds - nicely discriminative objects without any background, sampled with perfectly regular, even coverage and low, isotropic noise. If one claims practical utility, then one should nowadays really show at least "semi-real" point clouds (e.g., Scannet, semantic3d.net, KITTI, ...)


**Summary Of The Paper:**

The paper studies 3D point cloud data. The core message is that one can regard a point cloud as a collection  of samples from an underlying distribution and, hence, one can imagine a manifold of point clouds and construct an associated distance metric via the Fisher information. That distance can then be used for certain manipulation and analysis tasks, e.g., to interpolate ("morph") between point clouds along the manifold, or to regularise representation learning. A central assumption of the paper is that it is easier to choose a meaningful probability distribution than to make other a-priori assumptions "ad hoc" about the point cloud. However it is not discussed how to actually choose a suitable distribution, the paper limits itself to Gaussian kernel density (seemingly also "ad hoc" for mathematical convenience).


**Summary Of The Review:**

A rather theoretical paper, the proposed view is in some sense elegant, but not as revolutionary as claimed. Interesting theoretical or practical developments could perhaps build on the proposed view, although the paper does not push far enough in any directions to get to a real impact. Still, I have no objections against putting the idea out to the community, if the authors tone down their claims and acknowledge that they are throwing out a cute, raw idea; while neither exploring the theoretical implications in much depth nor engineering it into a strong algorithmic tool.

---

> ### Author Response · Authors · 2021-11-18
> **Response to Reviewer PyPd (2/2)**
>
> __Q2.__ The kernel density flavour suggests that the method might
> not scale all that well to realistic point cloud sizes. Perhaps it does,
> but the experimental setting with $< 2100$ points per object evades the topic
>
> __A2.__ Thank you for this constructive comment. As the reviewer pointed out,
> kernel density estimators may not be scalable for data such as LiDAR or
> Scannet. We mainly address object-level point cloud data in this paper, and
> each point of this data contains important information for recognizing the
> object (e.g., classification). However, for large scale data such as
> LiDAR and Scannet, one point of the point cloud data may not be as
> important as in the case of objects. In such cases, one could first fit
> the scene point cloud with GMM, and then apply our regularization algorithm.

---

> ### Author Response · Authors · 2021-11-18
> **Response to Reviewer PyPd (1/2)**
>
> Thank you very much for your constructive feedback, and our sincerest
> apologies for not uploading our comments sooner. In response to the
> many constructive suggestions we have received, we have spent the
> past week conducting a wide range of new experiments, and revising
> our manuscript accordingly. In an attempt to answer the reviewer
> questions, and to better clarify and validate our contributions,
> the appendix has been substantially expanded with the following
> additional content:
> * we have revised the introduction so that it introduces some related/existing works and explains what our contributions are in those contexts;
> * we have added additional results with SOTA autoencoder models in Table 1 in Section 5.2;
> * we have added experiments on ``noisy" point cloud data in Section 5.2, which demonstrates the advantages of our method more clearly;
> * we have added Appendix A (Existing Geometric/Statistical Methods for Point Cloud Data);
> * we have added some more details of algorithms in Appendix D;
> * we have added Appendix E.1.1 (Quantitative Results on Synthetic Data);
> * we have added Appendix E.2.1 (Performance Analysis with Varying Regularization Coefficients);
> * we have added Appendix E.2.2 (Learning Curves of Noisy Point Cloud Data);
> * we have added Appendix E.2.3 (Experimental Results on Semi-supervised Classification);
> * we have added Appendix F (Computational Aspect);
> * we have added Appendix G (Different Kernel Choices for Density Estimations).
>
> Below we provide detailed responses to each of the reviewer comments.
> When referencing any major changes and addition of new content to the
> revised manuscript, we have indicated those passages ``like this``.
>
> __Q1.__
> The reviewer's main concerns are as follows:
> * A central assumption of the paper is that it is easier to choose a meaningful probability distribution than to make other a-priori assumptions "ad hoc" about the point cloud. However it is not discussed how to actually choose a suitable distribution, the paper limits itself to Gaussian kernel density (seemingly also "ad hoc" for mathematical convenience);
> * The view as a distribution may be useful in some
> situations - typically when requiring a "rough, global" descriptor of
> the point cloud, for instance to recognise well-discriminated shape
> categories; it is almost certainly not a suitable view for many other
> (arguably more realistic and relevant) tasks - for instance the kernel
> smoothing will harm any local feature extraction or description; and it
> will likely also be challenged by point clouds that cover the object of
> interest incompletely or with varying density;
> * nothing is said to substantiate the claim that choosing
> the distribution is easier than making other reasonable assumptions;
> I am not sure how to make a reasonable choice for realistic distributions
> (say, autonomous vehicle Lidar).
>
> __A1.__ In the previous version of the paper, eliminating the ``ad hoc ness"
> of existing algorithms was described as a main contribution. In the revised
> paper, as suggested by several reviewers, we have re-positioned our paper's
> contributions in the context of existing related works.
>
> We have revised the introduction so that it includes
> existing related works, and explains what our contributions are in
> these contexts. Additionally, more detailed discussion on why our
> proposed method is novel vis-\`{a}-vis existing geometric/statistical
> methods has been added in Appendix A.  Our contributions can be
> summarized as follows: (i) we construct, in a mathematically
> rigorous way, the statistical manifold structure of point cloud data
> using a 1-1 mapping, and (ii) based on the constructed 1-1 mapping
> and properties of the statistical manifold, we construct a Riemannian
> manifold structure, including an analytic form of the Riemannian
> metric that is particularly amenable to practical implementation.
>
> The reviewer asked excellent and challenging questions:
> ``How to choose a suitable distribution (or kernel function in our
> context) in more realistic point cloud data that has some sharp local
> features, varying local density, sparse point samples, large number
> of point samples, and etc".  We agree that it is important to think
> about these and other complexities of real-world problems.
>
> To solve these real-world problems, we believe the problem
> should be divided into several well-posed sub-problems.
> For example, consider the case of road data obtained through lidar
> sensors of autonomous vehicles.  Point cloud data will include
> various objects such as cars, pedestrians, trees, buildings, and lanes.
> Of course, the local density, sharpness of the features, etc. of the
> point cloud data obtained from each object will be very different.
> We think that object detection should come first, and after that, we can use different
> kernel functions depending on the type of object. Moreover, when the local density is significantly different, using a variable bandwidth is expected to be a solution.

---

### Official Review · Reviewer_FLA8 · 2021-11-01

**Correctness:** 4
**Technical Novelty And Significance:** 3
**Empirical Novelty And Significance:** 3
**Recommendation:** 8
**Confidence:** 3

**Main Review:**

The idea of representing point clouds as probability distributions and considering those as points in a statistical manifold equipped with the Firsher information metric seems good and well-founded. I am not aware of any previous work that has exactly this take on point cloud analysis.

Overall, I find the idea novel, the presentation clear and interesting.

Some points that the authors can consider:

- a weakness of the method is the estimation of the density. The result is of course very dependent on the chosen kernel density estimator and parameters of the kernels. As far as I could read, the authors use fixed values for the bandwidth without justification for the chosen values. A more comprehensive discussion of the effect of the density estimator and e.g. the bandwidth parameter would strengthen the paper.

- I was at the first read confused about the statement that the mapping h:X\to S being 1-1, and subsequently that the representation is invariant to permutations. I believe the manifold \mathcal X can equivalently be considered the set of equivalence classes of point sets under the permutation group so that permutation invariance comes by construction, but perhaps this could be described more explicitly around Proposition 1.

- Proposition 3: Isn't the linear independence assumption satisfied on all reproducing kernel Hilbert spaces? Perhaps the result can be stated much more generally.

**Summary Of The Paper:**

The paper concerns data sets of point clouds in Euclidean space and proposes to analyze those as samples from underlying probability distributions. Each point cloud being represented by a probability distribution can now be seen as a point in a statistical manifold on which the Fisher information metric provides a natural Riemannian structure. The authors presents two experiments in this setup where they train autoencoders to represent point clouds, interpolate between point clouds using geodesics, and where the latent representation is trained so that straight line are close to geodesics.

**Summary Of The Review:**

I believe the paper presents a novel take on analysis of point clouds.

---

> ### Author Response · Authors · 2021-11-18
> **Response to Reviewer FLA8 (2/2)**
>
> __Q3.__ Isn't the linear independence assumption satisfied on all reproducing kernel Hilbert spaces? Perhaps the result can be stated much more generally.
>
> __A3.__ We have been searching literature on reproducing kernel Hilbert spaces
> and trying to find relevant theories or proofs, but we haven't been able
> to find them yet. We would greatly appreciate it if you could point us to
> where one might find such relevant information.

---

> > ### Comment · Reviewer_FLA8 · 2021-11-19
> > **Reponse to A3.**
> >
> > I believe the linear independence follows directly from invertibility of the matrix resulting from evaluating the kernel at any combination of distinct points and then direct computation.

---

> > > ### Author Response · Authors · 2021-11-23
> > > **Response about Linear Independence Condition**
> > >
> > > You are right, thank you very much for your insightful comment.
> > > As you said, it is true that "the linear independence follows directly from invertibility of the matrix resulting from evaluating the kernel at any combination of distinct points (i.e., strictly positive definite kernel)" [1, 2].
> > > Now, this gives an easier way to find kernels that satisfy the linear independence condition in Proposition 2.
> > > Gaussian kernel used in this paper is strictly positive definite, which automatically proves Proposition 3.
> > > Besides, other kernels that are strictly positive definite such as the Laplacian kernel and inverse multiquadratic kernel can be used to construct the statistical manifolds [3].
> > > We again appreciate the constructive comments; we will definitely include this discussion in the final revised version.
> > >
> > > [1] Paulsen, V. I., & Raghupathi, M. (2016). An introduction to the theory of reproducing kernel Hilbert spaces (Vol. 152). Cambridge university press.
> > >
> > > [2] Ferreira, J. C., & Ferreira, E. C. (2018). On Reproducing Kernel and Applications. Advances in Analysis, 3(1), 11-22.
> > >
> > > [3] Sriperumbudur, B. K., Gretton, A., Fukumizu, K., Schölkopf, B., & Lanckriet, G. R. (2010). Hilbert space embeddings and metrics on probability measures. The Journal of Machine Learning Research, 11, 1517-1561.

---

> ### Author Response · Authors · 2021-11-18
> **Response to Reviewer FLA8 (1/2)**
>
> Thank you very much for your constructive feedback, and our sincerest
> apologies for not uploading our comments sooner. In response to the
> many constructive suggestions we have received, we have spent the
> past week conducting a wide range of new experiments, and revising
> our manuscript accordingly. In an attempt to answer the reviewer
> questions, and to better clarify and validate our contributions,
> the appendix has been substantially expanded with the following
> additional content:
> * we have revised the introduction so that it introduces some related/existing works and explains what our contributions are in those contexts;
> * we have added additional results with SOTA autoencoder models in Table 1 in Section 5.2;
> * we have added experiments on ``noisy" point cloud data in Section 5.2, which demonstrates the advantages of our method more clearly;
> * we have added Appendix A (Existing Geometric/Statistical Methods for Point Cloud Data);
> * we have added some more details of algorithms in Appendix D;
> * we have added Appendix E.1.1 (Quantitative Results on Synthetic Data);
> * we have added Appendix E.2.1 (Performance Analysis with Varying Regularization Coefficients);
> * we have added Appendix E.2.2 (Learning Curves of Noisy Point Cloud Data);
> * we have added Appendix E.2.3 (Experimental Results on Semi-supervised Classification);
> * we have added Appendix F (Computational Aspect);
> * we have added Appendix G (Different Kernel Choices for Density Estimations).
>
> Below we provide detailed responses to each of the reviewer comments.
> When referencing any major changes and addition of new content to the
> revised manuscript, we have indicated those passages ``like this``.
>
> __Q1.__a weakness of the method is the estimation of the density. The result is of course very dependent on the chosen kernel density estimator and parameters of the kernels. As far as I could read, the authors use fixed values for the bandwidth without justification for the chosen values. A more comprehensive discussion of the effect of the density estimator and e.g. the bandwidth parameter would strengthen the paper.
>
> __A1.__ Thank you for your comment and suggestion. In the revised paper, ``
> we have added in Appendix G a brief qualitative analysis on how the
> statistical representation changes when a kernel function other than
> the Gaussian kernel function is used.``
>
> About the bandwidth parameters, we visualize the 3d kernel density function
> of the point cloud data according to its value (as shown in Figure 2), and
> choose the bandwidth value so that the density function seems to have
> a natural shape.  We agree that more comprehensive discussion of the
> effect of different bandwidth parameters is needed, and we will attempt
> to conduct additional autoencoder regularization experiments varying the
> bandwidth parameter of the Gaussian kernel.
>
> In Appendix G, we compare the Gaussian, Uniform, Triangular, Epanechnikov,
> and Sigmoid kernel functions; for better visualization, comparison is
> made with 2d point clouds: circles, ellipses, squares, stars, and hearts.
> In short, kernel functions with infinite support (e.g., Gaussian and sigmoid)
> produce smooth estimates, whereas ones with finite support (e.g., uniform,
> triangular, and Epanechnikov) produce some discrete behavior. The overall
> resulting shapes are similar, but each has its own characteristics, so we may
> select specific kernel functions according to the given point cloud data; we
> choose the one that produces the smoothest estimates, the Gaussian kernel.
> In future work, we will also examine which kernel function are reasonable choices (i.e.,
> satisfying Proposition 3), and how the type of kernel function
> affects the result of autoencoder regularization.
>
> __Q2.__ I was at the first read confused about the statement that the
> mapping $h:X \to S$ being 1-1, and subsequently that the representation is
> invariant to permutations. I believe the manifold $\mathcal{X}$ can equivalently
> be considered the set of equivalence classes of point sets under the permutation
> group so that permutation invariance comes by construction, but perhaps this
> could be described more explicitly around Proposition 1.
>
> __A2.__ Thank you for raising this point. Your understanding is correct, and we
> agree that the notation $\mathbf{X}$ and $\mathcal{X}$ can be made
> more explicit. ``We have therefore revised the
> notation as follows.`` When referring to the point cloud
> data $\mathbf{X}$, the equation has been modified as follows:
>
> Before:
> $\mathbf{X}=\{(x_1,...,x_n)\:
> | \: x_i\in\mathbb{R}^{D}, x_i\neq x_j \text{ if } i \neq j\}$
>
> After:
> $\mathbf{X}=\{x_1,...,x_n\:
> | \: x_i\in\mathbb{R}^{D}, x_i\neq x_j \text{ if } i \neq j\}$
>
> The point cloud data $\mathbf{X}$ is itself now the equivalence class
> of point sets under the permutation group, and the manifold $\mathcal{X}$
> is the set of equivalence classes of point sets. The 1-1 mapping
> $h:\mathcal{X} \to S$ is then defined in terms of this new notation.

---

### Official Review · Reviewer_21tj · 2021-11-02

**Correctness:** 4
**Technical Novelty And Significance:** 2
**Empirical Novelty And Significance:** 3
**Recommendation:** 3
**Confidence:** 5

**Main Review:**

Strength:
I personally find that incorporating the proposed constraint to achieve Fisher information metric in the underlying latent space is a very interesting idea.

As the authors proposed, the underlying latent space created by the autoencoder does seem to be endowed with the desired metric. The experiments regarding the decision boundary is also interesting, as the data seems to be cleanly separated.

The task of point cloud classification does not really involve usage of pointwise correspondence, therefore may have worked to the advantage of the authors' proposal.

Weakness:
The contribution of the paper is very limited, as it lacks a thorough review of the geometric/statistical representation of point clouds in other fields.
The description of representing points as a distribution is accurate, yet has already been explored in registration. The proposal in section 3 itself lacks novelty in that regard.

In the field of image/point cloud registration, there has already been decades of discussion regarding the usage of geometric and statistical representation, and the advantages of using Euclidean metric and Fisher information metric to compare the sets of points.
In the case of point set registration, it is widely known that geometric methods excel at cases where outliers are included, while statistical methods are robust to noise.

Such discussion has been completely omitted in this proposal, which I find disappointing.
Here are some of the prior works, but there are many more:

H. Chui and A. Rangarajan. "A new point matching algorithm for non-rigid registration," CVIU 2003

A. Myronenko and X. Song, “Point set registration: Coherent point drift,” TPAMI 2010

B. Jian and B. Vemuri, "Robust point set registration using gaussian mixture models," TPAMI 2010

Z. Zhou et al. “Robust non-rigid point set registration using student’s-t mixture model,” PloS one, 2014

There has also been discussion regarding merging the two approaches:

F. Li et al, "Toward a unified framework for point set registration," ICRA 2021

The novelty of the paper lies in section 4, which I did find interesting. However, as it lacks a thorough review, the contribution of this section is insufficient.
For example, as stated above, each method is known to have some advantages and disadvantages in registration. I wonder if, in this proposal, the same can be said of cases under a different setting, such as point clouds with added noise, occlusion, etc.
The overall novelty lies only in the proposal of embedding the metric in the latent space, therefore thorough analysis is definitely necessary for its contribution to be sufficient.

**Summary Of The Paper:**

This paper proposes to represent point cloud data as samples drawn from statistical distribution, and project the data to an underlying statistical manifold on which various analyses can be conducted in a manner that is more favorable than doing so in a regular Euclidean space.

The authors thoroughly describe how points are converted into samples from a probability density function, and explain the metric that determines the coordinate system on the statistical manifold, which describe the changes to the distribution of point clouds.
To introduce this metric to represent point clouds, the authors propose to use this to find minimal geodesic curve between two data in a latent space. They also propose to add a novel constraint to the error metric of autoencoders, so that the encoded latent space possesses the Fisher information metric.

The effect of this proposal is firstly analyzed by two synthetic experiments. First, the authors show that using the proposal, they are able to find the geodesic curve along the manifold that consists of data with the same label. Then, they demonstrate that the latent space of the constrained autoencoder successfully captures the Riemann metric, as samples of many classes are placed along a single line, or are cleanly separated.
Finally, the authors transfer the learnt representation between dataset to demonstrate that the proposed metric space is suitable for representation of various point cloud data.

**Summary Of The Review:**

Overall, I find the paper to be lacking background analysis, and thus also in novelty.

The autoencoder with the Fisher information metric as the metric of the latent space is an intriguing proposal. However, as this is the only contribution of the paper, the authors are required to thoroughly compare the geometric and statistical approaches. There has been decades of research that analyzed the advantages of the two representations, especially in the field of image and point set registration. Therefore, such knowledge had to be used in the analysis. As the paper lacks such analysis, I have to lean towards rejection.

---

> ### Author Response · Authors · 2021-11-18
> **Response to Reviewer 21tj (2/2)**
>
> __Q2.__ The novelty of the paper lies in section 4, which I did find interesting. However, as it lacks a thorough review, the contribution of this section is insufficient. For example, as stated above, each method is known to have some advantages and disadvantages in registration. I wonder if, in this proposal, the same can be said of cases under a different setting, such as point clouds with added noise, occlusion, etc. The overall novelty lies only in the proposal of embedding the metric in the latent space, therefore thorough analysis is definitely necessary for its contribution to be sufficient.
>
> __A2.__ As mentioned in __A1__, we believe the novelty of our
> paper lies not only in the proposed latent space metric, but also a
> rigorous construction of the point cloud Riemannian manifold -- we do
> believe this is important and not just an inconsequential intellectual
> exercise -- and development of an explicit expression of the
> Riemannian metric that is amenable for practical applications.
>
> That being said, we fully agree with the reviewer that a more thorough
> and comprehensive analysis with further autoencoder experiments will
> make our contributions clearer.  We have therefore carried out additional
> experiments, and added more analysis to the main manuscript and Appendix.
> Specifically, ``(i) we have added a more thorough analysis of
> experimental results in Appendix E.1.1 (quantitative analysis on
> synthetic dataset) and Appendix E.2.1 (performance analysis on
> standard benchmark dataset), (ii) we have added experimental results
> for noisy point cloud data in Section 5.2 and Appendix E.2.2,
> and (iii) we have added experimental results
> for semi-supervised classification setting Appendix E.2.3,``
>
> We believe the added contents should better demonstrate the
> significance and pratical value of our proposed method.
>
> Among the newly added contents, we would like to share the following
> result: we have conducted additional experiments to determine, given
> noisy point cloud data, how much more robust is the representation obtained with our regularization approach. We add different levels of
> noise (1\%, 5\%, 10\%, and 20\%) to each point cloud data, and compare the classification accuracy for a vanilla autoencoder and a regularized
> autoencoder. Details of the experiments can be found in Appendix D.4
> (Details for Experiments on Standard Benchmark Dataset with Noise).
> The results are summarized in the table below:
>
> |                      |        |  MN  |  40     |       |       |   MN    |  10     |       |
> |------------------|-------|--------|--------|-------|-------|-------|-------|-------|
> | Method           | 1%    | 5%    | 10%   | 20%   | 1%    | 5%    | 10%   | 20%   |
> | FcNet            | 87.8% | 83.2% | 75.6% | 64.5% | 92.4% | 91.9% | 88.4% | 79.8% |
> | FcNet + E (ours) | 86.6% | 85.1% | 79.1% | 70.4% | 92.2% | 91.1% | 88.2% | 82.6% |
> | __FcNet + I (ours)__ | __89.0%__ | __86.6%__ | __81.4%__ | __72.4%__ | __93.3%__ | __92.6%__ | __91.6%__ | __84.8%__ |
>
> Compared to vanilla autoencoders (FcNet), regularized autoencoders
> (FcNet + I) show much higher classification accuracy overall, with
> the gap in classification accuracy becoming even more pronounced with
> increasing noise levels.  Another interesting finding concerns the
> learning curve (see Appendix E.2.2): when the noise level is set to
> 10\% and 20\%, the classification accuracy of the vanilla autoencoder
> gradually decreases as learning progresses (as the epoch increases),
> but such a phenomenon does not occur in the regularized autoencoder.
> We refer to Appendix E.2.2 (Learning Curves for Noisy Point Cloud Data)
> for the additional information.
>
> Moreover, our regularized autoencoders (i.e., FcNet + I) also show higher classification accuracy in semi-supervised setting, and the gap in classification accuracy becomes even more pronounced with decreasing number of labeled training data.
> Please refer to Appendix E.2.3 (Experimental Results on Semi-supervised
> Classification) for details of experimental setting and experimental results including the learning curves.
>
> |                      |        |  MN  |  40     |       |       |   MN    |  10     |       |
> |------------------|-------|--------|--------|-------|-------|-------|-------|-------|
> | Method                       | 50%    | 10%    | 5%   | 1%   | 50%    | 10%    | 5%   | 1%   |
> | FcNet                         | 85.7% | 78.0% | 70.6% | 50.3% | 91.7% | 90.1% | 87.2% | 74.1% |
> | __FcNet + I (ours)__ | __87.9%__ | __81.6%__ | __76.8%__ | __57.4%__ | __93.2%__ | __91.2%__ | __88.3%__ | __78.1%__ |
>
> Thank you once again for your comments, we found them very
> constructive and helpful in improving our paper.

---

> ### Author Response · Authors · 2021-11-18
> **Response to Reviewer 21tj (1/2)**
>
> Thank you very much for your constructive feedback, and our sincerest
> apologies for not uploading our comments sooner. In response to the
> many constructive suggestions we have received, we have spent the
> past week conducting a wide range of new experiments, and revising
> our manuscript accordingly. In an attempt to answer the reviewer
> questions, and to better clarify and validate our contributions,
> the appendix has been substantially expanded with the following
> additional content:
>
> * we have revised the introduction so that it introduces some related/existing works and explains what our contributions are in those contexts;
> * we have added additional results with SOTA autoencoder models in Table 1 in Section 5.2;
> * we have added experiments on ``noisy" point cloud data in Section 5.2, which demonstrates the advantages of our method more clearly;
> * we have added Appendix A (Existing Geometric/Statistical Methods for Point Cloud Data);
> * we have added some more details of algorithms in Appendix D;
> * we have added Appendix E.1.1 (Quantitative Results on Synthetic Data);
> * we have added Appendix E.2.1 (Performance Analysis with Varying Regularization Coefficients);
> * we have added Appendix E.2.2 (Learning Curves of Noisy Point Cloud Data);
> * we have added Appendix E.2.3 (Experimental Results on Semi-supervised Classification);
> * we have added Appendix F (Computational Aspect);
> * we have added Appendix G (Different Kernel Choices for Density Estimations).
>
> Below we provide detailed responses to each of the reviewer comments.
> When referencing any major changes and addition of new content to the
> revised manuscript, we have indicated those passages ``like this``.
>
> __Q1.__ The contribution of the paper is very limited, as it lacks a thorough review of the geometric/statistical representation of point clouds in other fields. The proposal in section 3 itself lacks novelty in that regard.
> Such discussion has been completely omitted in this proposal, which I find disappointing.
>
> __A1.__
> Thank you very much for your many suggested references; we fully agree with your comment that our review of related studies was insufficient. We
> have accordingly ``revised the introduction to include more
> existing related works, and to place our contribution in the context
> of this existing body of work. We have also added more detailed discussion
> in Appendix A on the novelty and research significance of our method
> compared to existing geometric and statistical methods for point cloud
> analysis.``
>
> After a more careful examination of existing related works, we agree
> with the reviewer that the statistical interpretation of point cloud data
> has already been addressed in numerous previous works. That being said,
> we still believe there is clear novelty and impact in our approach,
> and we have tried to better clarify what these are in the revised version.
> Here is a summary of our contributions: (i) we construct, in a mathematically
> rigorous way, the statistical manifold structure of point cloud data
> using a 1-1 mapping, and (ii) based on the constructed 1-1 mapping
> and properties of the statistical manifold, we construct a Riemannian
> manifold structure, including an analytic form of the Riemannian
> metric that is particularly amenable to practical implementation.
>
> In Appendix A, we provide a thorough review of existing geometric
> and statistical representations of point cloud data. Briefly, this section includes a "geometric methods'' section (A.1) and a
> "statistical methods'' section (A.2). In the section on geometric methods,
> we review some popular distance metrics developed for point cloud data,
> and explain why distance metrics alone is insufficient for many
> practical applications. This in turn allows us to highlight the
> importance of the overall Riemannian manifold framework that we
> develop in this paper.  In the section on statistical methods,
> we review existing statistical representations that have been used
> for tasks ranging from point cloud registration to denoising, highlighting
> some of the gaps in previous studies and how these gaps are filled
> by our method.
>
> We hope that the revised introduction and added appendix better clarify
> the research significance and impact of our contribution, not only
> from a theoretical perspective, but also for a wide range of practical
> applications.

---

### Decision · Program_Chairs · 2022-01-20

**Decision:**

Reject

**Comment:**

This paper proposes to interpret point cloud data in Euclidean space as samples from some underlying probability distribution. Thus a set of point cloud data can be given the structure of a statistical manifold with a Riemannian metric structure, namely the Fisher Information Metric.
Applications to point cloud autoencoders are then studied.

Reviewers generally agree that the idea of equipping point cloud data with the Fisher information metric is interesting and has potential. However, there are concerns that the theoretical properties of the proposed framework have not been explored in depth. Furthermore, the experimental work should be enhanced to demonstrate the practical utility of the proposed formulation.